**1** **Impact of LUCC on Streamflow Based on the SWAT Model over the**

**2** **Wei River Basin on the Loess Plateau of China**

Hong Wang[1], Fubao Sun*[1,2,3,5], Jun Xia[4,5], Wenbin Liu[1]
[1] Key Laboratory of Water Cycle and Related Land Surface Processes, Institute of
Geographic Sciences and Natural Resources Research, Chinese Academy of Sciences,
Beijing 100101, China
[2] Research School of Qilian Mountain Ecology, Hexi University, Zhangye City, Gansu
Province, 734000, China
[3] College of Resources and Environment, University of Chinese Academy of Sciences,
Beijing, 100049, China
[4] State Key Laboratory of Water Resources and Hydropower Engineering Sciences,
Wuhan University, Wuhan, 430072, China
[5] Center for Water Resources Research, Chinese Academy of Sciences, Beijing
100101, China
*Corresponding author: Fubao Sun (sunfb@igsnrr.ac.cn)

**Abstract:** Under the Grain for Green project in China, vegetation recovery constructions have been widely implemented on the Loess Plateau for the purpose of soil and water conservation. Now it becomes controversial whether the recovery constructions of vegetation, particularly forest, is reducing streamflow in rivers of the Yellow River Basin. In this study, we choose the Wei River, the largest branch of the Yellow River and implemented with revegetation constructions, as the study area. To do that, we apply the widely used Soil and Water Assessment Tool (SWAT) model for the upper and middle reaches of the - Wei River basin. The SWAT model was forced with daily observed meteorological forcings (1960-2009), calibrated against daily streamflow for 1960-1969, validated for the period of 1970-1979 and used for analysis for 1980-2009. To investigate the impact of the LUCC (Land Use and land Cover Change) on the streamflow, we firstly use two observed land use maps of 1980 and 2005 that are based on national land survey statistics emerged with satellite observations. We found that the mean streamflow generated by using the 2005 land use map decreased in comparison with that using the 1980 one, with the same meteorological forcings. Of particular interest here, we found the streamflow decreased in agricultural land but increased in forest area. More specifically, the surface runoff, soil flow and baseflow all decreased in agricultural land, while the soil flow and baseflow of forest were increased. To investigate that, we then designed five scenarios including (S1) the present land use (1980), (S2) 10%, (S3) 20%, (S4) 40% and (S5) 100% of agricultural land was converted into mixed forest. We found that the streamflow consistently increased with agricultural land converted into forest by about 7.4 mm per 10%. Our modeling results suggest that forest recovery constructions have positive impact on both soil flow and base flow compensating reduced surface runoff, which leads to a slight increase in streamflow in the Wei River with mixed landscapes of Loess Plateau and earth-rock mountain.

# 1. Introduction

Since 1999, China's Grain for Green project has greatly increased the vegetation cover (Chen et al., 2015) and the total conversion area reaches 29.9 million ha until 2014 (Li, 2015). And the proposals are to further return another 2.83 million ha farmland to forest and grassland by 2020 (NDRC, 2014). The establishment of either forest or grassland on degraded cropland has been proposed as an effective approach to mitigating climate change because these types of land use can increase soil carbon stocks (Yan et al., 2012; Deng et al., 2013). Implementation of large scalar Grain for Green project is undoubtedly one type of geoengineering which not only mitigates climate change but also is expected to alter hydrological cycle (Lacombe et al., 2016; Lacombe et al., 2008).

Some researchers have urged a cessation on Grain for Green expansion on the Loess Plateau of China and argued that continued expansion of revegetation would cause more harm than good to communities and the environment (Chen et al., 2015). One important reason was that the Grain for Green project lead to annual streamflow of the Yellow River declining (Chen et al., 2015; Li, 2001). Land use change can disrupt the surface water balance and the partitioning of precipitation into evapotranspiration, runoff, and groundwater flow (Sriwongsitanon and Taesombat, 2011; Foley et al., 2005; Wagner et al., 2013). Large scale revegetation constructions change hydrologic cycle process and distribution of water resources. There are three controversial points of view about the impact of vegetation on streamflow as a whole. Quite a few catchment studies indicated that annual streamflow decreased with revegetation increasing (Zhang and Hiscock, 2010; Bosch and Hewlett, 1982; VanShaar et al., 2002; Mango et al., 2011; Farley et al., 2005; Liu and Zhong, 1978) or increased with vegetation destruction ( Bosch and Hewlett, 1982; Woodward et al., 2014;

Hibbert, 2001), where some catchment studies indicated baseflow of forests was lower due to their
high evapotranspiration rates (Lørup et al., 1998; Lorup and Hansen, 1997; Smith and Scott, 1992),
while other studies indicated the baseflow increased in the dry season due to higher infiltration
and recharge of subsurface storage (the "sponge-effect hypothesis") (Price, 2011; Lørup et al.,
1998; Ogden et al., 2013). In contrast, other studies showed that vegetation has a positive impact
on streamflow (Tobella et al., 2014; Li et al., 2001) or no impact on streamflow (Wang, 2000;
Beck et al., 2013).

To interpret the controversial results, it was argued that the impact of vegetation on annual

streamflow depends on watershed area and the relationship between them was negative in smaller
watershed and positive in larger watershed (Huang et al., 2009; Zhang, 1984). Some of them
thought it was probably the large amount of transpiration water played the main function in
hydrological process when the watershed was smaller. And some thought that the different impacts
of area probably because the forest of larger watershed could increase precipitation and vegetation
was also conducive for the infiltration of precipitation, which increased the proportion of the
underground flow of streamflow in forest region. Some researchers indicated tree planting has
both negative and positive effects on water resources and the overall effect was the result of a
balance between them, which were strongly dependant on tree density (Tobella et al., 2014).
Lacombe et al. (2016) found soil infiltrability was an important factor for explaining two modes of
afforestation (natural regeneration vs. planting) led to opposite changes in streamflow regime.
Huang (1982) analyzed Soviet research results found that 48% runoff coefficients increased, 32%
has no change, and 20% decreased with watershed forest increasing. The increased regions were
located at high latitude and humid areas. Under this condition, the total evaporation in wooded
areas and woodless area are equal. The speculation was that snow may be blown away or to
wooded areas from woodless area, which could enhance the coefficient of streamflow but these
factors would be weaker over low to middle latitude than that in high latitude (Huang, 1982).
Further, vegetation may change hydrological cycle as follows (Le Maitre et al., 1999): redirection
of precipitation by the canopy; branches, stem and litter tends to intercept more water into the soil;
roots may provide channels for the flow infiltrating to groundwater and extract soil water as
evaporation. Hence different results have led to contentious relationship between vegetation and
streamflow (Bradshaw et al., 2007; Dijk et al., 2009).
The Wei River is one main branch of the Yellow River and has been widely implemented
measures of soil and water conservation, including forestation, terraces, grass and check dam,
since the 1980s. Meanwhile the annual streamflow of the Wei River has decreased significantly
since the 1980s (Liu and Hu, 2006; Lin and Li, 2010; Wang et al., 2011). Since the 1990s, the
streamflow has sharply dropped and the observed streamflow of Linjiacun station in the 1990s was
less than one third of that before 1990s. The terrace and check dam both had a negative effect on
annual streamflow which was a result of the balance between the streamflow reducing in the flood
season and baseflow increasing in non-flood season on the Loess Plateau (Shao et al., 2013a; Xu
et al., 2012). But the impacts of vegetation on streamflow are controversial and complicated.
Meanwhile on the Loess Plateau, it was found that there is a drying layer of soil underneath forest
with a depth of over 1 m to 3 m from the soil surface owing to serious soil desiccation in
water-limited ecosystems (Li, 2001; Wang, 2010a). The land use, rainfall, soil type and slope
gradient had a significant impact on dried soil layer thickness (Wang, 2010b). And the great water
deficit prevents gravitational infiltration of rainfall and replenishment of groundwater. So forests
on the Loess Plateau reduced streamflow as the results of increased retention of rainfall and
reduced recharge into ground water (Li, 2001; Tian, 2010). But for earth-rock mountain landscape,
vegetation grows on thinner soil layer of rock mountain, which is apt to be saturated and produce
soil flow on relatively impermeable rock. So the streamflow in wooded areas might be larger than
that in adjacent woodless areas. Under this situation, forests may have positive impact for
producing streamflow (Liu and Zhong, 1978).
To investigate that, we develop hydrological experiments based on the widely used SWAT
model and observed hydrological/ meteorological data and land use data in the Wei River. We aim
at understanding possible impact of revegetation constructions, especially the forest restoration on
streamflow and its components in the Wei River, which is not only the largest branch of the
Yellow river but also with very mixed landscape with the loess plateau and earth-rock mountain.
In Sect. 2, we describe the study area and data. In Sect. 3, we set up, calibrate, and validate the
SWAT model in the Wei River. Section 4 reports the numerical experiment results, which is then
followed by the conclusion in Sect. 5.
## 2. Study area and data
## 2.1 Study area
Wei River is the largest tributary of the Yellow River, which originates from the north of the
Wushu Mountain at an altitude of 3495 m (involving Gansu, Ningxia and Shaanxi Provinces), and
runs across 818 km through into the Yellow River at Tongguan County, Shaanxi Province. In this
study, we choose the basin of the upper and middle reaches ($4.68 \times 10^4 \, km^2$) of the Wei River basin
($103.97° \sim 108.75°$ E, $33.69° \sim 36.20°$ N, $13.48 \times 10^4 \, km^2$). And the Linjiacun, Weijiabu and
Xianyang hydrological stations are used from upstream to midstream in this study (Fig. 1), which
divided the study area into 3 regions. Linjiacun station locates at the control section of the
upstream and Xianyang station is the control station of middle reaches.
Geologically, the basin consists of the Loess Plateau and Qinling Mountain in the respective
north and south of the Wei River (Fig. 1). In the north, there are fewer tributaries, whose lengths
are further and the gradient is smaller. While in the south, abundant tributaries originate from
Qinling Mountain which is steep and close to the river. So the tributaries are shorter and the flows
are swifter. And there distribute lots of earth-rock mountain landscape and gravel riverbed in the
piedmont.

## 141 2.2 Land Use and land Cover Change (LUCC) data

We obtained observed LUCC data from National Science & Technology Infrastructure of
China, National Earth System Science Data Sharing Infrastructure (Fig.2) (http://www.geodata.cn).
Land use maps for the years of 1980 and 2005 were interpreted based on the corresponding
national land use survey data (1:100,000), satellite image, the MODIS data, 250-meter space
resolution data and combined with pasture resources map (1:500,000), soil type map (1:1,000,000),
vegetation type map (1:1,000,000) and other auxiliary data. The LUCC data were divided into six
types and further 25 subtypes. And the six types included forest, shrubland, pasture, cropland,
water bodies and residential areas: ①  The forest type includes Range-Brush (RNGB),
Forest-Mixed (FRST), Forest-Deciduous (FRSD), Pine (PINE) and Forest-Evergreen (FRSE); ②
The pasture type includes Pasture (PAST), Winter Pasture (WPAS) and Range-Grasses (RNGE);
③ The cropland means Agricultural Land (AGRL); ④ Water includes water (WATR) and
Wetlands-Mixed (WETL); ⑤ The residential areas include area of Residential-High Density
(URHD) and Residential-Medium Density (URMD); ⑥ The code of bare type is BARE. The
area of agricultural land decreased about 7.26% and forest area increased 0.81% in 2005 compared
with 1980 for the study area.

## 2.3 Soil data

Soil data were obtained from National Science & Technology Infrastructure of China,

National Earth System Science Data Sharing Infrastructure (Fig. 3(a)) (http://www.geodata.cn).
This soil data map reflects the distribution and characteristics of different soil type and digitized
based on 1:500,000 remote sensing digital figures of environment on Loess Plateau.

Based on the soil data, the distribution of earth-rock mountain in study area is drawn as Fig.

3(b). There were 83 soil types in the study area and 15 of them are composed of earth and rock
involving 70 hydrological response units (HRUs) (Table 1). At the same time, these 15 soil types
distribute mainly in the Qinling Mountain and Liupan Mountain (Fig. 1). And the earth-rock
mountain area accounts for 24% of study area.

## 2.4 Meteorological and hydrological data

The meteorological data were obtained from the China Meteorological Data Sharing Service

System (http://www.escience.gov.cn/metdata/page/index.html) and some local rainfall stations.
The data include atmospheric pressure, mean (minimum and maximum) temperature, vapor
pressure, relative humidity, rainfall, wind speed, wind direction, sunshine time. Figure 4 (a) shows
the distribution of meteorological stations and the annual average precipitation over Wei River
basin, which was calculated using kriging interpolation method of ArcGIS 9.3 based on annual
average precipitation of 34 meteorological stations. Then the time series of annual average
precipitation for the three regions of the study area were calculated respectively using elevation
bands method of ArcSWAT (Soil and Water Assessment Tool) 2009.93.7b, which can account for
orographic effects on precipitation (Neitsch et al., 2011). SWAT allows the subbasin to be split
into a maximum of ten elevation bands. Precipitation is calculated for each elevation band as a
function of the respective lapse rate and the difference between the gage elevation and the average
elevation specified for the band. Once the precipitation values have been calculated for each
elevation band in the subbasin, new average subbasin precipitation value is calculated based on
the fraction of subbasin area within the elevation band (Neitsch et al., 2011). Figure 5 (b), (c) and
(d) show the time-series of average precipitation calculated though elevation bands method of
ArcSWAT from 1960 to 2009. The average of precipitation of region 1, 2 and 3 were 489.71
493.25 and 566.60 mm/yr and the trend analysis showed that the precipitation of them decreased
with an average decreasing rate of 0.57, 0.55 and 0.21 mm/yr, whereas the decreasing tendencies
were not significant at the 0.05 level.

And the daily streamflow data of three hydrological stations were obtained from Ecological

Environment Database of Loess Plateau (http://www.loess.csdb.cn/pdmp/index.action) and the
Hydrological Year books of China. Figure 5 (b), (c) and (d) show the time-series of annual
streamflow and runoff coefficients in the three regions of study area. The trend analysis showed
that streamflow of region 1 and 2 decreased extremely significantly (P < 0.01), with an average
decreasing rate of 1.74 and 5.38 mm/yr. The streamflow of region 3 did not decreased significantly.
And the average runoff coefficients were 0.13, 0.34 and 0.17 in region 1, 2 and 3 over the past 50
years (1960-2009). The trend analysis of runoff coefficients showed that the tendencies of region 1
and 2 decreased extremely significantly (P < 0.01), with an average decreasing rate of 0.34%, and
1.09 % per year. The runoff coefficient of region 3 decreased significantly (P < 0.01) too, with an
average decreasing rate of 0.2% per year.
90-meter resolution digital elevation model (DEM) (Fig. 4 (b)) was used to define the
topography characteristics (such as elevation, slope and aspect) and delineate the watershed
boundary. It was obtained from the Computer Network Information Center, Chinese Academy of
Sciences (http://srtm.datamirror.csdb.cn/), based on the Shuttle Radar Topography Mission (SRTM)
version 4.1.

## 3. Methods

### 3.1 The SWAT model

The SWAT model is developed by the USDA Agricultural Research Service (ARS). It is a
physically based and distributed hydrological model. The SWAT model has been widely applied to
understand the impact of land management practices on water, sediment and agricultural yields
over large complex watersheds with varying soils, land use and management conditions over long
periods (Arnold et al., 2009). It is forced with meteorological data, and input with soil properties,
topography, land use, and land management practices in the catchment. The physical processes
associated with hydrological cycle and sediment movement etc. are directly modeled by SWAT
using these input data (Arnold et al., 2009). In addition, the ArcSWAT extension (ArcSWAT
2009.93.7b version) is used as the graphical user interface for the SWAT model (Gassman et al.,
2007; Arnold et al., 1998). For the streamflow, surface runoff, soil and baseflow are considered.
Soil flow is streamflow contribution which originates below the surface but above the zone where
rocks are saturated with water. Base flow is the volume of streamflow originates from
groundwater (Arnold et al. 1993).

### 3.2 The SWAT Model setup

The SWAT model setup includes four steps: watershed delineation, hydrological response
unit (HRU) analyst, input database building and modification and model operation. Based on
research of the Wei River (Shao, 2013b; Wang, 2013), the extraction threshold, which is the
minimum drainage area required to form the origin of a stream, of subbasin area was 80 km$^2$. The
Linjiacun, Weijiabu and Xianyang hydrological stations were loaded manually as subbasin outlets
and one whole watershed outlet was defined. The study area was divided into 308 subbasins (Fig.
1). The land area in a subbasin can be further divided into the HRUs, which is the basic computing
element of the SWAT model. In this study, a subbasin was subdivided into only one HRU that was
characterized by dominant land use and soil type. Then the daily meteorological data, including
temperature, relative humidity, sunshine duration, wind speed, rainfall, were input and all data
were written into database building and modification to force the SWAT model.

For evaluating the performance in the model calibration and validation, we use the R$^2$ and NS

coefficient to evaluate the performance rating of the model (Nash and Sutcliffe, 1970) (Equation
(1) & (2)).
$$R^2 = \frac{\left[\sum_{i=1}^{n}\left(O_i^{obs} - \overline{O_i^{obs}}\right)\left(O_i^{sim} - \overline{O_i^{sim}}\right)\right]^2}{\sum_{i=1}^{n}\left(O_i^{obs} - \overline{O_i^{obs}}\right)^2\left(O_i^{sim} - \overline{O_i^{sim}}\right)^2}$$
Eq. (1)

$$NS = 1 - \frac{\sum_{i=1}^{n}\left(O_i^{obs} - O_i^{sim}\right)^2}{\sum_{i=1}^{n}\left(O_i^{obs} - \overline{O_i^{obs}}\right)^2}$$
Eq. (2)

where n is the number of observations, $O^{obs}$ is the observed value, $O^{sim}$ is the simulated value, and
the overbar means the average of the variable. The R$^2$ describes the proportion of the variance in
measured data explained by the model and typically 0.5 is considered an acceptable threshold
(Santhi et al., 2001; Van Liew and Garbrecht, 2003). The SWAT model simulation can be judged
as "satisfactory" if the NS > 0.50 for a monthly time step simulation and the performance rating of
the SWAT model was very good when the NS > 0.75, and the model performed good when the
NS > 0.65 (Moriasi et al., 2007).

## 3.3 Calibration and validation of the SWAT model

We setup the SWAT-CUP procedure for the sensitivity analysis, calibration and validation in
our study (Abbaspour, 2007). The sensitivity analysis is carried out by keeping all parameters
constant to realistic values, while varying each parameter within the range assigned in step one.
The sensitive parameters were calibrated using LH-OAT (Latin-Hypercube-One Factor-At-a-Time)
method of the Sequential Uncertainty Fitting (SUFI2) program (Abbaspour, 2007; Xu et al., 2012).
And the t-stat and p-value were used to evaluate the sensitivity of parameters. The t-stat is the
coefficient of a parameter divided by its standard error and the larger values are more sensitive.
And the p-value determines the significance of the sensitivity and a value close to zero means
more significant. The most sensitive (seven) parameters were selected by the SWAT-CUP module.
Combined with previous research in Wei River, two additional parameters (SOL_K and
GW_DELAY) with the seven parameters were selected in this study (Table 2).
The initial value and the range of relevant parameters were derived from previous research in
study area (Wang, 2014; Shao, 2013b; Zuo et al., 2015). Vegetation construction changes
undelaying surface and affects quantity of surface runoff and recharge of both soil and ground
water. It has a significant impact on infiltration by providing canopy and litter cover to protect the
soil surface from raindrop impacts and producing organic matter which can bind soil particles and
increase soil porosity (Le Maitre et al., 1999). These impacts of vegetation on hydrological
process are epitomized and reflect by CN and management operation in the SWAT model. the Soil
Conservation Service (SCS) curve number equation is the model for computing the amounts of
streamflow in SWAT model and its comprehensive parameter is CN which relates to the soil's
permeability, land use and antecedent soil water conditions. We have done some research on the
impacts of LUCC changes on runoff, infiltration and groundwater under different soil, slope and
rainfall intensity in Wei River basin based on simulated rainfall experiments before (Wang, 2014).
Based on the experiments, the SCS model and the three-dimensional finite-difference groundwater
flow model (MODFLOW) were calibrated and applied also. So values of parameters related to
runoff, infiltration and groundwater, such as the initial CN values and recharge rates for different
LUCC, specific yield of soil layer etc. were gotten based on experiments and mathematical
simulation (Wang, 2014). Meanwhile in the SWAT model, agricultural land and forest have
different heat units required for plant maturity and different management operations. The
agricultural land includes plant, harvest/ kill and auto-fertilizer operation and the forest only has
plant operation. And the management operation of forest involves leaf area index (LAT_INIT),
plant biomass (BIO_INIT), age of trees (CURYR_MAT).

The revegetation was mainly implemented in the study area after the 1980s. Hence we

choose 1960-1969 and 1970-1979 for the model calibration and validation respectively and used
the daily streamflow data of the Linjiacun, the Weijiabu and the Xianyang hydrological stations
from the upper to middle reaches (the data of 1965 and 1968-1971 are missing in the Weijiabu
station). The parameters were calibrated for hydrological stations by the order of upstream to
midstream using the daily streamflow of 1960-1969. Firstly, the parameters against the streamflow
at the Linjiacun control station were calibrated. Secondly, based on the premise of the calibrated
parameter values of the Linjiacun station, the parameters were calibrated for the subbasin
controlled by the Weijiabu station. In that way, the parameters for the subbasin controlled by the
Xianyang station were then calibrated. Then the SWAT model was validated for the three
hydrological stations respectively against the streamflow from 1970 to 1979 (Fig. 6).

## 4. Results and discussions

The corresponding statistic results of three hydrological stations showed that the ranges of
NS and $R^2$ were 0.59~0.66 and 0.63~0.68 respectively in the calibration period for a daily time
step. And they were 0.57~0.62 and 0.61~0.65 respectively in the validation period. At a monthly
time step, the results of the NS and $R^2$ were 0.82~0.84 and 0.79~0.86 respectively in the
calibration period. And they were 0.70~0.76 and 0.74~0.79 respectively in the validation period
demonstrating good performance of the model. In addition, the time-series and the patterns of the
simulated and observed streamflow during the calibration and validation periods showed similar
trends (Fig. 6). Our conclusion is that the SWAT model can be used in upper and middle reaches
of the Wei River basin.

## 4.1 Impact of the observed LUCC on streamflow

Analysis above (Fig. 5) showed that the observed precipitation of study area did not
decreased significantly from 1960 to 2009, while the annual streamflow (except region 3) and
runoff coefficients decreased significantly ($P < 0.05$) under this meteorological conation. This
discrepancy could attribute to LUCC changes mostly (Lacombe et al., 2016; Lacombe et al., 2008).
In order to analyze the impact of the LUCC on streamflow, the land use data of the 1980 and 2005
were used in the validated SWAT model and the DEM and soil data remained constant. Firstly,
the daily streamflow from 1980 to 2009 were simulated using observed daily meteorological
forcing data and topography, soil data in study area. Secondly, the LUCC data of 1980 was
replaced by that of 2005 and their relevant parameters of corresponding land use type were also
replaced. We used the LUCC data of 2005 but the same meteorological data to simulate the daily
streamflow from 1980 to 2009.

The change of annual streamflow based on LUCC data of 2005 compared with LUCC data of

1980 showed that annual streamflow of Xianyang hydrological station decreased during 20-year in
30-year ((1980-2009)) and the annual average reduction was 2.0 mm/yr for these 20-year in study
area. This result was consistent with the decreasing tendencies of the observed streamflow of
Xianyang station, which decreased significantly ($P < 0.05$), with an average decreasing rate of
2.45 mm/yr from 1980 to 2009. The modelled streamflow represent the impacts of constant LUCC
data of 1980 and 2005, whereas observations are based on dynamic LUCC data, which could
explain the discrepancy. Yin et al. (2017) studied the impact of LUCC changes on streamflow in
Jinghe River basin, which is the largest tributary of the Wei River basin, found that the streamflow
increasingly influenced by LUCC changes, which contributed to 44% of the streamflow changes
between the 1980s and 1990s and 71% of the streamflow changes between the 1990s and 2000s.
At the same time, different land use types hydrological responds differently even to the same
meteorological forcings, i.e., rainfall intensity was of great importance influencing to hydrological
process in semi-dry and semi-humid region (Lacombe et al., 2008; Wang, 2014). Results of
rainfall experiments showed when the rainfall intensity was smaller or larger, the rainfall would
infiltrate into soil or flow away as surface runoff mainly on both grass land and bare slope, while
when the rainfall intensity was medium, the rainfall would infiltrate into grass land and flowed
away as surface runoff on bare slope (Tobella et al., 2014; Wang, 2014). To reduce influence of
meteorological conditions and isolate the impact of the LUCC on streamflow, the 30-year
(1980-2009) values of the streamflow for forest and agricultural land were averaged respectively.
For period of 1980-2009, we just used their measured and long-term daily meteorological data in
the study area to drive the validated model for the designed hydrological experiments. Figure 7
shows the changes of streamflow, surface runoff, soil flow and baseflow between agricultural land
and forest. The surface runoff, soil flow and baseflow all decreased for agricultural land, while the
soil flow and baseflow of forest increased. Overall, the streamflow decreased in agricultural land
and increased in forest area. When the LUCC data are classified and re classified in SWAT model,
the tree types are summarized as Range-Brush (RNGB), Forest-Mixed (FRST) and
Forest-Deciduous (FRSD). Different types have different hydrological responses for their leaf,
roots and so on. We also analyzed the streamflow generation of the main types of forest (RNGB,
FRST and FRSD) in study area further. Results showed that the streamflow yield of FRST and
FRSD were about 1.20 and 1.60 times of that of RNGB respectively.

## 340   4.2 Hydrological experiments on the impact of conversion of

## 341   agricultural land to forests on streamflow

Because the LUCC data involves various land use interconversions, of particular interest here
the impact of conversion of cropland to forest on streamflow cannot be distinguished. Starting
from the LUCC data of 1980 as (S1) the present land use, we design other four scenarios (Table 3)
that (S2) 10%, (S3) 20%, (S4) 40% and (S5) 100% of the agricultural land was converted into
Forest-Mixed (FRST) respectively. And all experiments carried out based on the same the DEM,
soil data and meteorological conditions.
Based on the five scenarios, the SWAT simulations were conducted to analyze the effect of
forest constructions on the streamflow in upper and middle reaches of the Wei River basin. Firstly,
the converted agricultural land area was controlled proportionately as same as the variational area
ratios of set scenarios in 3 regions divided by Linjiacun, Weijiabu and Xianyang hydrological
stations (Fig. 5(a)). Secondly, lands with the same soil type and similar slope were the priorities
choosing as the converted land. Thirdly, the converted lands were distributed evenly as much as
possible in 3 regions. The simulation period was from 1980 to 2009.
We present the distribution of average streamflow change under S2 ~ S5 scenarios compared
with S1 scenario in Fig. 8. It shows that the streamflow generally increased when the land use
converted from agricultural land into forest in the upstream. And Fig. 9 shows the change rate of
streamflow at the Linjiacun, Weijiabu and Xianyang stations correspondingly for its annual
average and annual average over non-flood season (Jan - Jun and Nov - Dec). Compared with the
S1 scenario, the annual average streamflow increases in the non-flood season were 12.70 %,
11.21 % and 9.11% for the Linjiacun, Weijiabu and Xianyang stations with per 10% area of
agricultural land converted into forest. Interestingly the average annual streamflow increases were
11.61%, 21.63%, 42.51% and 109.25% for S2, S3, S4 and S5 scenario respectively (Fig. 9 (b)),
which almost consistently suggested about 1.1% per 1% change of the agricultural land. The
results are important in that one can expect that for a 0.8% increase in the forest in the observed
LUCC would lead to less than 1% change in the streamflow, which is negligible.
To be more comparable, Fig. 10 show the distribution of the annual runoff coefficients with
the scenario changed from S1 to S5. The spatial variability in mean runoff coefficient was large,
which ranges from 0.03 to 0.68 and increased with more forest converted from agricultural land.
The annual average runoff coefficient of study area increased from 0.21 to 0.37 with forest area
increasing from S1 to S5 (Fig. 11). On average, the runoff coefficient increased about 0.014 (i.e.,
1.4% of rainfall transformed into streamflow) with per 10% area of agricultural land converted
into forest.
The landscape of the Wei River is mixed with the Loess Plateau and earth-rock mountain
landscapes, which induce different mechanisms of transforming rainfall into streamflow. The
earth-rock mountain area accounts for 24.03% of study area (Fig. 3 (b)). In earth-rock mountain
area, vegetation grows on much thinner soil layer over the earth-rock mountain. And the soil has
high infiltration ability for high stone fragment content. The thin soil is apt to be saturated and
produce more soil flow on relatively impermeable rock, hence the streamflow in wooded areas is
larger than that in adjacent woodless areas favoring streamflow production (Liu and Zhong, 1978).
On the contrary, in Loess Plateau there is exiting a drying layer of soil underneath forestland in
great water deficit. When the agricultural land converted into forest, the precipitation, intercepted
by vegetation, infiltrated into soil and supplied the drying layer of soil, vegetation growth, etc.
Together with much thicker soil layer on the Loess Plateau, it usually prevents gravitational
infiltration into groundwater and reduces streamflow recharge (Li, 2001; Tian, 2010). The
observed results of precipitation and streamflow in study area also showed the runoff coefficients
had obviously positive correlation with rates of earth-rock mountain area. The regional annual
averages of runoff coefficient were 0.13, 0.17 and 0.34 for Fig. 5 (b), (d) and (c), while the rates of
earth-rock mountain area were opposite correspondingly (Fig. 3 (b)). The complication is that the
overall effect of forest on the streamflow is in fact a balance between earth-rock mountain positive
and Loess Plateau negative effects on the streamflow.
Combined with the spatial distribution of precipitation (Fig. 4 (a)), we can see earth-rock
mountain landscapes are mainly distributed in regions with more rainfall. To be precise, the whole
earth-rock mountain area located where rainfall was greater than 500 mm/yr and over 62% of the
study area where the annual rainfall is greater than 600 mm was in earth-rock mountain.
Meanwhile, the river network over the earth-rock mountain is denser and most of tributaries in the
earth-rock mountain are close to the main stream of the Wei River. Moreover, there distribute a lot
of developed gravel riverbed in piedmont, sandy soil along the river and its groundwater level is
shallow, which facilitate rainfall infiltration and recharging streamflow. Therefore although the
area of earth-rock mountain accounts for 24% of the study area, its distribution areas are
concentrated in the main regions of streamflow yield of the study area. Therefore the overall result
of balance among all factors was that the forest constructions have a little positive effect on
streamflow in study area.
Seemingly, this result was not consistent with the significant decreasing tendencies of
streamflow in study area. The combined effects of LUCC, including forestation, terraces, grass,
and dam, could explain the discrepancy. Under the same meteorological conditions, the
streamflow is mainly a result of combined effects of these measures. Results showed the terrace in
the main Weihe River basin could delay the flood and add the drought season streamflow, which
reduced the annual streamflow in general. The terrace in 2000 could decrease about 37 million $m^3$
annual water and increased the most dry month streamflow by 3.5% in Xianyang station (*Shao,*
*2013b)*. Zhang et al (*2014a, 2014b*) studied the terrace measures of Yanhe River basin, typical
basin of the Loess Plateau, and results showed that the terrace measures could reduce the runoff in
the flood season and increased the baseflow. Results showed that 1 $m^3$ water could be supplied to
the river when 5~ 6 $m^3$ water stored by the terrace. This meant the water reducing effect of terrace
was larger than 80% in Yanhe River basin and. Xu et al. (2012) applied the SWAT model to
simulate the streamflow in the Yanhe basin and results showed that the check dams had a
regulation effect on streamflow. From 1984 to 1987, the streamflow in rainy season (from May to
October) decreased by 1.54 $m^3s^{-1}$ (14.7 %) to 3.13 $m^3s^{-1}$ (25.9 %) due to the check dams; while in
dry season (from November to the following April), streamflow increased by 1.46 $m^3s^{-1}$ (60.5%)
to 1.95 $m^3s^{-1}$ (101.2 %); From 2006 to 2008, the streamflow in rainy season decreased by 0.79
$m^3s^{-1}$ (15.5 %) to 1.75 $m^3s^{-1}$ (28.9 %), and the streamflow in dry season increased by 0.51 $m^3s^{-1}$
(20.1 %) to 0.97 $m^3s^{-1}$ (46.4 %). Lots of results showed that the terrace and check dam both had a
negative effect on annual streamflow which was a result of the balance between the streamflow
reducing in the flood season and baseflow increasing in non-flood season on the Loess Plateau
(Shao, et al., 2012, 2013a, 2013b; Zhang, et al., 2014a, 2014b; Xu, et al., 2012). The observed
streamflow was a result of the balance among forestation, terraces, grass, and dam.

## 4.3 Impact of conversion of agricultural land to forests on baseflow

In Fig. 9 (a), one important point is that the average increase in the non-flood season was
about 1.41 times larger than the annual increase of the streamflow. To understand that, Fig. 12
shows distribution of the baseflow index, i.e., the ratio between baseflow and streamflow, under
S1~S5 scenarios. We can see that the baseflow index also increased with land use converted from
agricultural land into forest, which means that groundwater contribution to the streamflow
increased with the overall increase of forest area. Putting the pictures together, Fig. 13 shows the
changes of the streamflow and the baseflow under the S2~S5 scenarios minus those results under
the S1 scenario in the non-flood season. The average increases of streamflow and baseflow were
1.14 and 0.98 mm/yr with per 1% increase of forest area respectively. For the non-flood season,
they were 0.60 and 0.53 mm/yr. The increase of the streamflow contributed by the increased
baseflow was about 88.33% in the non-flood season. So the increasing streamflow was mainly
contributed by groundwater with increasing of forest area overall.
Although some researchers have urged a cessation on Grain for Green expansion on the
Loess Plateau of China for it lead to annual streamflow of the Yellow River declining (Chen et al.,
2015; Li, 2001), our modeling results suggest that forest recovery constructions have a little
positive impact on both soil flow and base flow compensating reduced surface runoff, which leads
to a slight increase in streamflow in the Wei River with mixed landscapes of Loess Plateau and
earth-rock mountain. And rainfall patter also has great effect on streamflow, particularly the
extremes rainfall, i.e., Lacombe et al. (2008) found no streamflow change was found for when the
precipitation was larger than 40 mm. Results showed that the daily precipitation extremes seem to
be consistent with the 7% increase per degree of warming (Allen and Ingram, 2002; Pall et al.,
2007) and one-hour precipitation extremes increase twice as fast with rising temperatures as
expected when daily mean temperatures exceed 12 $^{\circ}$C (Lenderink and Meijgaard, 2008; Westra,
2014). The streamflow is the combined effects of LUCC (forestation, terraces, grass, and dam and
so on) and climate changes. The impact of Grain for Green project on streamflow should be
thoughtfully studied according to the characteristics of the basin.
At the same time, there are some uncertainties in SWAT model simulations. First, the SWAT
model could offer the comprehensive parameters for subbasin and detailed parameters for
different HRU according to their slopes, soli type and LUCC. The comprehensive parameters were
calibrated according to observed streamflow of subbasin, while the different parameters of HRU
could not be calibrated individually. Second, the model could not tell the impact of short-duration
rainfall on streamflow which has great effect on streamflow. In addition, watershed size,
generalization and data accuracy all can lead to uncertainty in the simulations ( Yin et al., 2017).
To reduce the uncertainty of simulation influence, the 30-year (1980-2009) values of streamflow
were averaged to analyze the impacts.

# 5. Conclusion


The large scalar implementation of Grain for Green project in China is expected to alter

hydrological cycle, in particular on the Loess Plateau, within the Yellow River Basin. The
scientific question is how large the impact of the LUCC on the streamflow and its components in
that area. We choose the Wei River as the study area, in that it has been widely implemented
revegetation constructions since the 1980s. Of particular interest here, the landscape of the upper
and middle reaches of the Wei River basin is mixed with the Loess Plateau and rocky mountain,
which would induce different mechanisms of generating surface runoff, soil flow, base flow and
therefore streamflow.

To investigate it, we setup the SWAT model for the upper and middle reaches of the Wei

River basin with the inputs of long term observed meteorological forcing data, hydrological data,
and observed land use data. We use daily and monthly streamflow of the Linjiacun, Weijiabu and
Xianyang hydrological stations from upper to middle reaches during 1960-1969 and 1970-1979
respectively for the model calibration and model validation. The results showed that the
Nash-Sutcliffe (NS) coefficients and the coefficients of determination ($R^2$) were > 0.57 and 0.61
for daily streamflow and 0.70 and 0.74 for monthly streamflow respectively demonstrating that
the SWAT model can be used in this study.

We analyze the impact of the LUCC on streamflow based on the observed LUCC data of

1980 and 2005. The daily streamflow from 1980 to 2009 were simulated using observed daily
meteorological data with the two different land use data. The results showed that two-thirds of
annual streamflow decreased and the change of streamflow was different among different land use.
On the overall average, the 30-year averages of the streamflow decreased in agricultural land but
increased in forest. To interpret the overall result, we design five scenarios in this study including
(S1) the present land use of 1980 and the scenarios where agricultural land was converted into
forest by 10% (S2), 20% (S3), 40% (S4) and 100% (S5) respectively. Based on the five scenarios,
we use the calibrated and validated SWAT model to analyze the effect of forest constructions on
the streamflow in detail. The results confirm that annual streamflow consistently increased with
more forest converted from the agricultural land. Interestingly, the rate is almost consistently 7.41
mm/yr per 10% increase of forest converted from the agricultural land. Based on detailed analysis
of each component of streamflow, we found it was most attributed by the baseflow. The overall
effect of LUCC on the streamflow in the Wei River basin, the largest branch of the Yellow River is
the result of the balance between Loess Plateau negative and earth-rock mountain positive effects.
Our results here are not only of great importance in understanding the impact of LUCC on
streamflow for a catchment with much complicated and mixed landscape, but also of significance
for water resources managing practice.
## Data availability
The data used in this manuscript were obtained from reliable public data repositories. The
LUCC and soil data were obtained from the National Science & Technology Infrastructure of
China, the National Earth System Science Data Sharing Infrastructure (http://www.geodata.cn).
The DEM data were obtained from the Computer Network Information Center, the Chinese
Academy of Sciences (http://srtm.datamirror.csdb.cn/). The meteorological data were obtained
from       the       China       Meteorological       Data       Sharing       Service       System
(http://www.escience.gov.cn/metdata/page/index.html). The daily streamflow data were from the
Ecological Environment Database of Loess Plateau (http://www.loess.csdb.cn/pdmp/index.action)
and the Hydrological Year books of China.

# Acknowledgment

This research was supported by the National Key Research and Development Program of

China (2016YFA0602402), an Open Research Fund of State Key Laboratory of Desert and Oasis
Ecology, Xinjiang, Institute of Ecology and Geography, Chinese Academy of Sciences,
CPSF-CAS Joint Foundation for Excellent Postdoctoral Fellows, National Key Research and
Development Program of China (2016YFC0401401), the Chinese Academy of Sciences (CAS)
Pioneer Hundred Talents Program, the International Science and Technology Cooperation
Program of China (2014DFA71910), Natural Science Foundation of China (41571028 and
41601035). We thank the Editor and reviewers for valuable comments that improved the
manuscript.

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

**Figure Captions:**
**Fig.1** The study area: the Wei river basin on the Loess Plateau.
**Fig. 2** The observed land use data of the year 1980 and the year 2005 in study area.
**Fig. 3** The Soil data and the distribution of earth-rock mountain in study area.
**Fig. 4** The spatial distribution of annual average precipitation in Wei River basin over the past 55 years
(1956-2010) and the DEM of study area.
**Fig. 5** The time-series of precipitation, annual streamflow and runoff coefficients for the regions of
study area.
**Fig.6** The time-series graphs of calculated vs. observed values during calibration period and
verification period for hydrological stations.
**Fig. 7** The changes of 30-year (1980-2009) averages of streamflow, surface runoff, soil flow and
baseflow between agricultural land and forest.
**Fig. 8** The watershed distribution of average streamflow change under S2~S5 scenarios compared with
S1 scenario.
**Fig. 9** The corresponding proportional change rate of streamflow at Linjiacun, Weijiabu and Xianyang
station for annual average and annual average in non-flood season.
**Fig. 10** The distribution of annual runoff coefficient with the scenario changed from S1 to S5.
**Fig. 11** The annual average runoff coefficient of study area with forest area increasing from S1 to S5.
**Fig. 12** The distribution of baseflow index under S1~S5 scenarios.
**Fig. 13** The corresponding change of streamflow and baseflow under S2~S5 scenarios compared with
S1 for annual average of year and non-flood season.

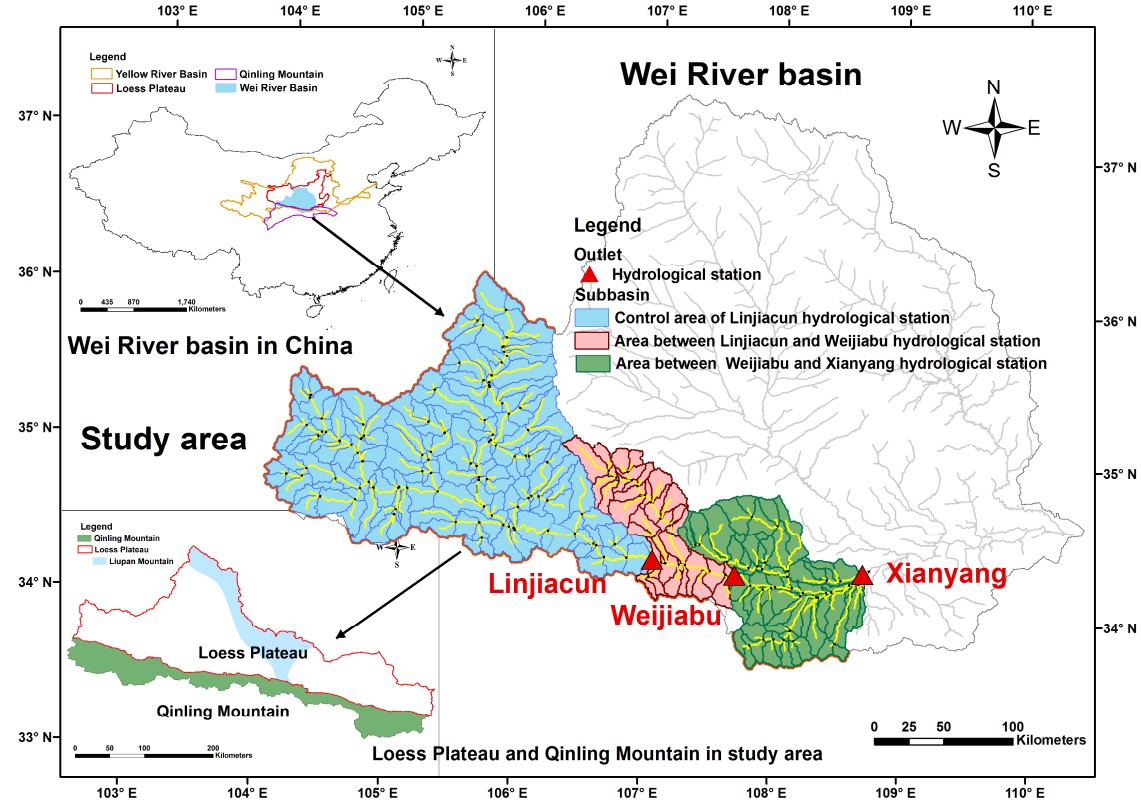

Fig. 1 The study area: the Wei river basin on the Loess Plateau.

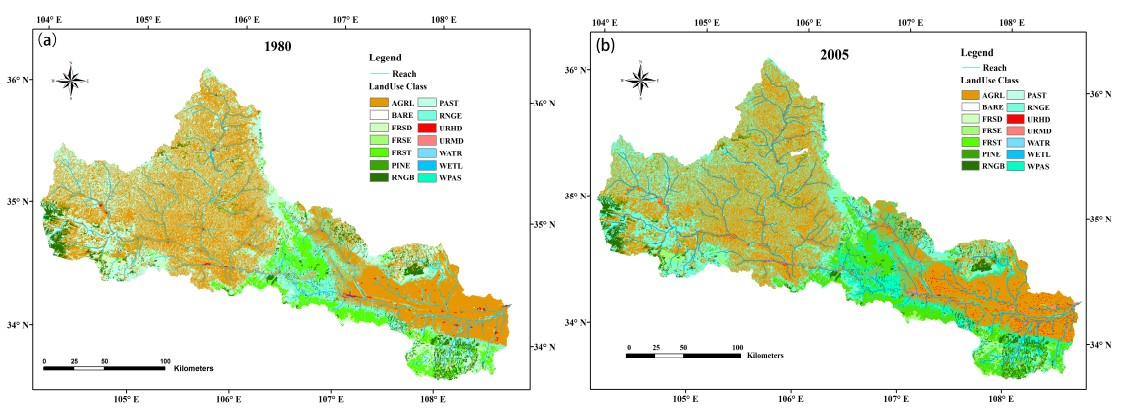

Fig. 2 The observed land use data of the year 1980 and the year 2005 in study area

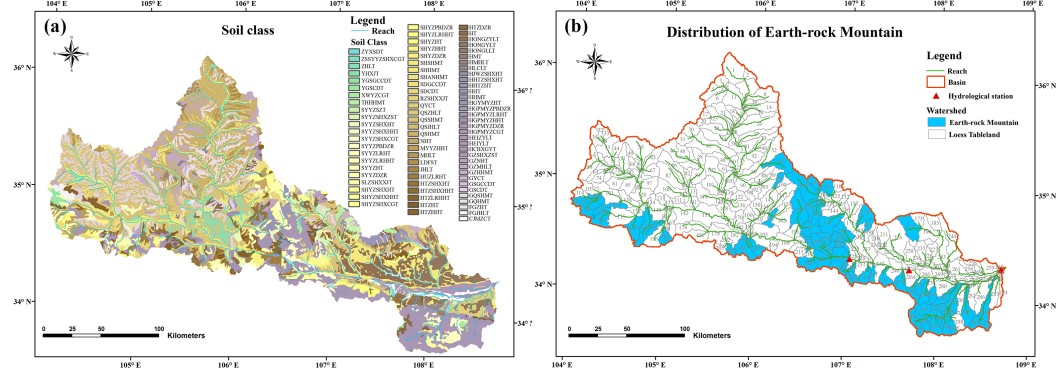

Fig. 3 The Soil data and the distribution of earth-rock mountain in study area


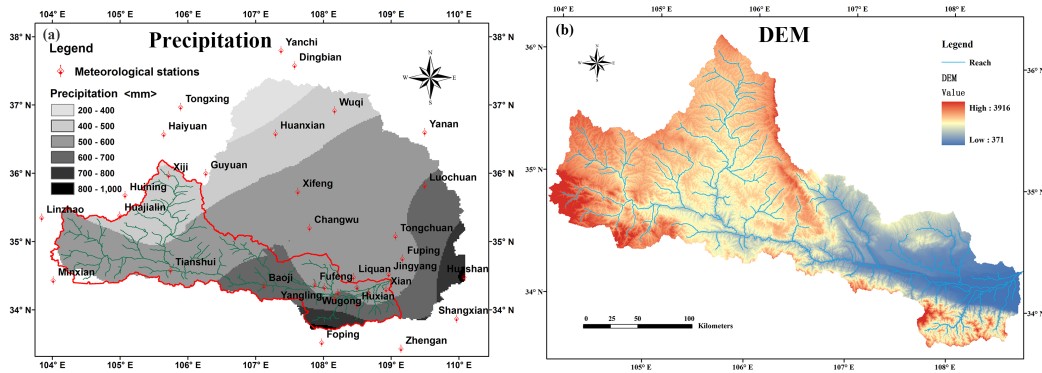

**Fig. 4** The spatial distribution of annual average precipitation in Wei River basin over the past 55 years

(1956-2010) and the DEM study area

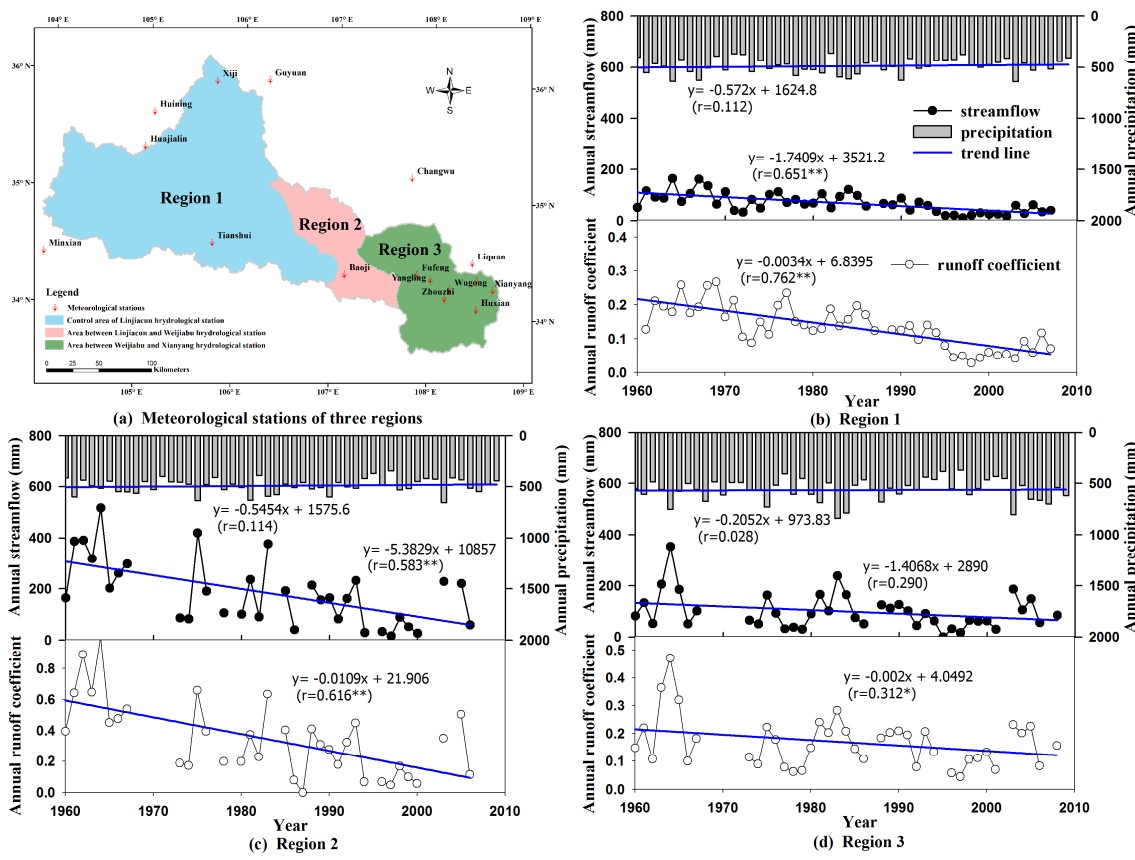


**Fig. 5** The time-series of precipitation, annual streamflow and runoff coefficients for the regions of

study area

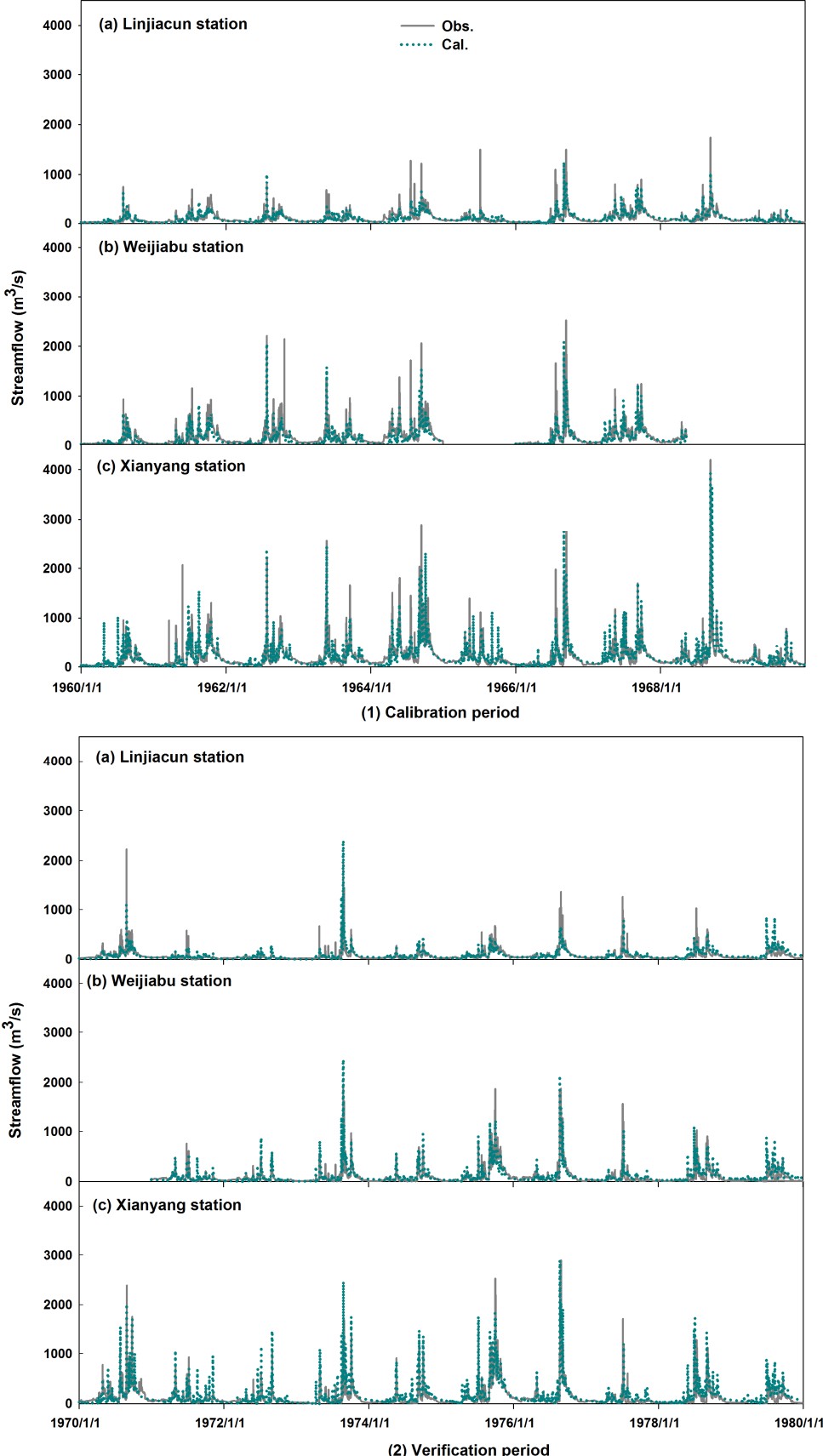


**Fig. 6** The time-series graphs of calculated vs. observed values during calibration period and verification

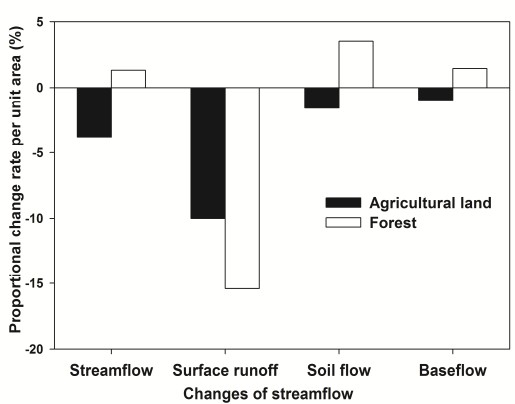


**Fig. 7** The changes of 30-year (1980-2009) averages of streamflow, surface runoff, soil flow and

baseflow between agricultural land and forest

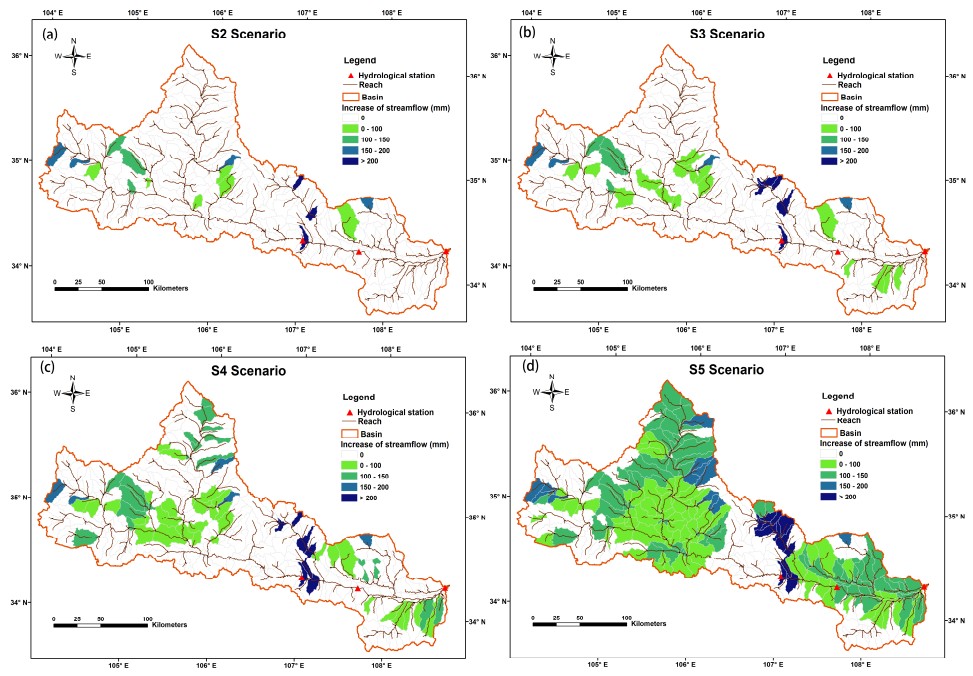


**Fig. 8** The watershed distribution of average streamflow change under S2~S5 scenarios compared with

S1 scenario

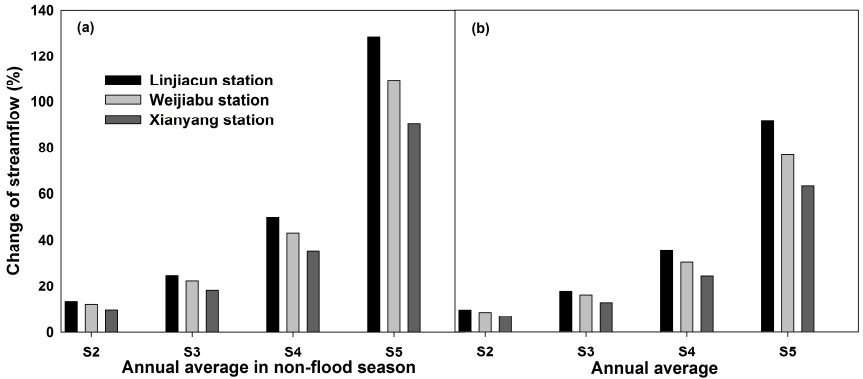


**Fig.9** The corresponding proportional change rate of streamflow at Linjiacun, Weijiabu and Xianyang

station for annual average and annual average in non-flood season


**Fig. 10** The distribution of annual runoff coefficient with the scenario changed from S1 to S5

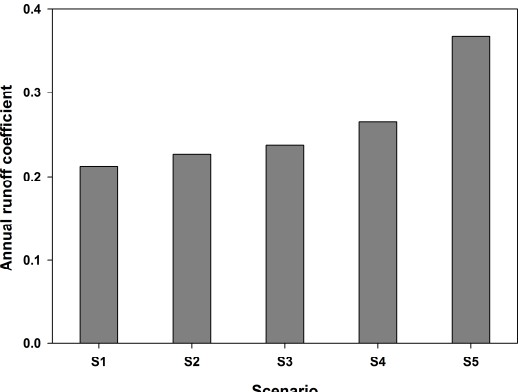


**Fig. 11** The annual average runoff coefficient of study area with forest area increasing from S1 to S5

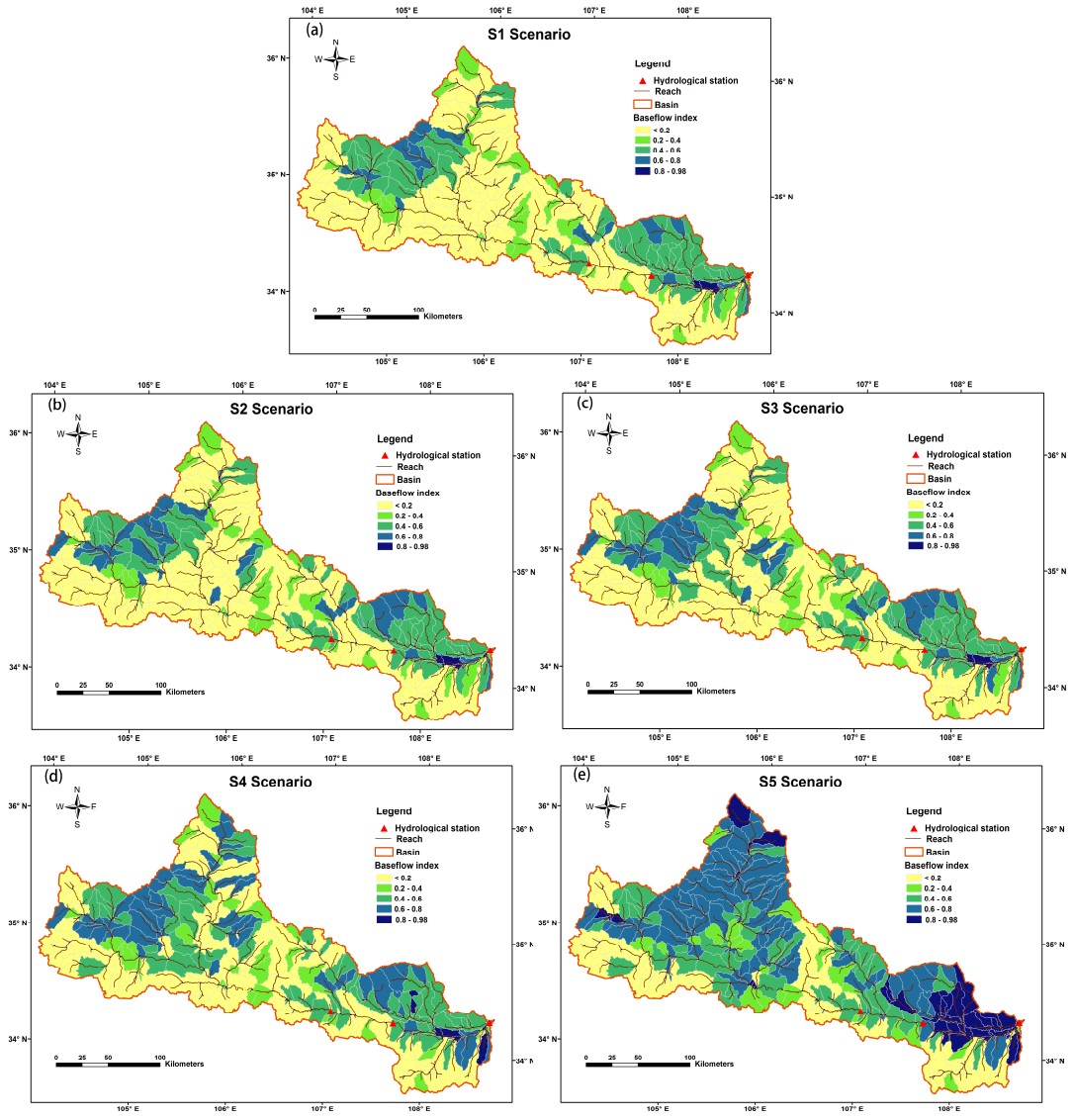


**Fig. 12** The distribution of baseflow index under S1~S5 scenarios

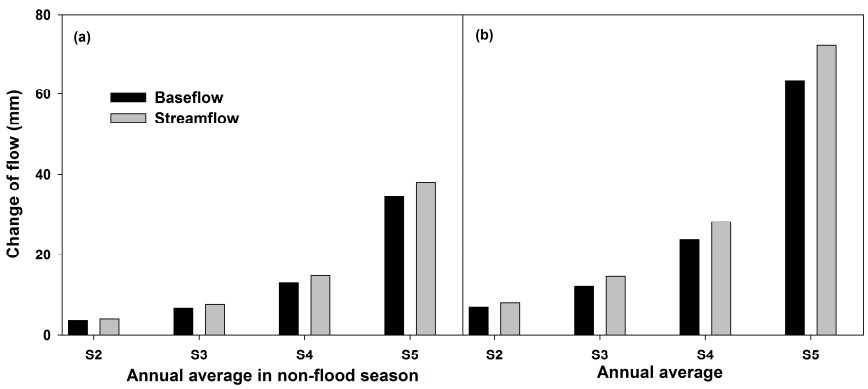


**Fig.13**The corresponding change of streamflow and baseflow under S2~S5 scenarios compared with
S1 for annual average of year and non-flood season
# Tables
Table 1 The soil type and its distribution of earth-rock mountain in study area

| No. | Code of Soil type | Physical meaning of the code | HRU | Area (km$^2$) |
|---|---|---|---|---|
| 1 | SHYZHT | Limestone Cinnamon soil | 220, 257 | 26316.90 |
| 2 | SHYZSHXHT | Limestone Calcic cinnamon soil | 153 | 11471.22 |
| 3 | SYYZLRHT | Sandstone—shale Luvie cinnamon soil | 166, 203, 207 | 50065.29 |
| 4 | HGPMYZLRHT | Granite—gneiss Luvie cinnamon soil | 174, 180, 187, 201, 204, 221, 277, 283, 287 | 158397.93 |
| 5 | SYYZDZR | Sandstone—shale Light brown earth | 106, 169, 299 | 103955.40 |
| 6 | HGPMYZDZR | Granite—gneiss Light brown earth | 130, 148, 172, 209, 252, 284, 289, 290, 291, 293, 294, 300, 301, 302, 303, 305, 306, 307, 308 | 299737.26 |
| 7 | HGPMYZPBDZR | Granite—gneiss Light brown earth | 253 | 8739.90 |
| 8 | MYYZHHT | Sandstone—shale Grey cinnamon soil | 115, 117, 146, 163 | 51204.96 |
| 9 | SYYZSHXHHT | Sandstone—shale Calcic grey cinnamon soil | 99, 129 | 19392.21 |
| 10 | SHYZSHXHHT | Limestone Calcic Grey cinnamon soil | 56 | 33885.54 |
| 11 | SYYZSHXZST | Sandstone—shale Purple soil | 109, 176, 177, 184, 200 | 106159.41 |
| 12 | HGPMYZCGT | Granit—gneiss Rhogosol | 165, 230, 237, 254, 271, 292, 295, 296, 297, 304 | 112136.40 |
| 13 | SYYZSHXCGT | Sandstone—shale Rhogosol | 107, 208, 213, 216, 218, 219, 248 | 87612.84 |
| 14 | SHYZSHXCGT | Limestone Rhogosol | 222 | 23375.79 |
| 15 | SYYZLRHHT | Sandstone—shale Luvic grey | 116, 140 | 30320.73 |

| | | cinnamon soil | | |
|---|---|---|---|---|


Table 2 Calibrated values of model parameters

| Parameters | Physical meaning | Calibration range | Calibration result | | |
|---|---|---|---|---|---|
| | | | Linjiac un | Weijia bu | Xianya ng |
| r__CN2 | Initial SCS runoff curve number for moisture condition Ⅱ | -0.3~0.3 | -0.27 | 0.05 | -0.17 |
| r__SOL_AWC | Available water capacity of soil layer | -0.6~0.6 | 0.01 | -0.01 | -0.01 |
| r__SOL_K | Saturated hydraulic conductivity of soil layer (mm/hr) | -0.5~0.5 | 0.5 | 0.3 | 0.5 |
| r__HRU_SLP | Average slope stepness (m/m) | -0.5~1.5 | 1.5 | 0.41 | 0.52 |
| r__SLSUBBSN | Average slope length (m) | -0.5~1.5 | 1.17 | 0.70 | 1.20 |
| v__ALPHA_BF | Baseflow alpha factor | 0~1.0 | 0.48 | 0.61 | 0.61 |
| v__GW_DELAY | Groundwater delay (days) | 0~500 | 220 | 38 | 62 |
| v__ESCO | Soil evaporation compensation factor | 0~1.0 | 0.65 | 0.90 | 0.80 |
| v__CH_K2 | Effective hydraulic conductivity in main channel alluvium | 0~130 | 5 | 30 | 30 |

Notes: v__ means the existing parameter value is to be replaced by the given value; r__ means the existing parameter value is multiplied
by (1+ a given value).

Table 3 Scenarios for simulation

| Scenario | Description | Area (km$^2$) | The average simulated streamflow (1980-2009) ($10^8$ m$^3$/yr) |
|---|---|---|---|
| S 1 | present situation | 0 | 50.44 |

| | | | |
|---|---|---|---|
| S 2 | 10% agricultural land → forest | 2937.63 | 53.92 |
| S 3 | 20% agricultural land → forest | 5875.26 | 56.83 |
| S 4 | 40% agricultural land → forest | 11750.53 | 62.73 |
| S 5 | 100% agricultural land → forest | 29376.32 | 82.28 |

Notes: ① Agricultural land refers to the land for crops planting, including cultivated land, newly cultivated soil, fallow field, rotation plot, pasture-crop rotation and land used for agro-fruit, agro-mulberry, agroforestry (The code in model is AGRL). ② Forest refers to the natural forest and plantation, which canopy density is larger than 30%, including timberland, economic forest, protection forest (The code in model is FRST).