# Peer review of "Impact of LUCC on Streamflow Based on the SWAT Model over the 2 Wei River Basin on the Loess Plateau of China"

_Hydrology and Earth System Sciences, 2016_

## Referee Comment (RC1) · G Lacombe (Referee) · 21 Aug 2016

This paper examines the hydrological effect of land conversion from agricultural land to forest in a sub-catchment of the Yellow River. The SWAT model is calibrated with daily hydro-meteorological data over the period 1960-1969 and validated over the period 1970-1979, using the same set of parameter values characterizing soils and land-use in 1980. Two land-use maps of the catchment are available for this study: one corresponding to the year 1980 (used for the model calibration) and one for the 2005 (used for simulation).

The calibrated SWAT model is then run twice to simulate flow using 1980-2009 meteorological input, the first time with land-use parameters of year 1980, and the second

time with land-use parameters of years 2005. The hydrological impact of land-use changes that occurred between 1980 and 2005 is quantified by comparing the two simulated flow time series. Finally, five land-use scenarios are defined, corresponding to a gradual increase in the percentage of agricultural lands converted to forest, and their effects on flow are simulated with the SWAT model.

Main comments:

This is an interesting topic but the approach needs significant improvement in order to provide evidence of the actual hydrological effects of the land-use and land-cover changes that occurred in the catchment. I recommend major revisions.

First of all, I am questioning the significance of the hydrological changes that actually occurred in the catchment over the studied period. Although figure 1 indicates that forested areas increased by about 65.104 hm2 (unusual unit used on the Y axis), which is equivalent to 14% of the upper catchment area, figure 3 inconsistently shows that forested area increased by only 0.81% (line 137) over the same period (1980 to 2005). How to explain this difference? If we rely on figure 2 (which is likely the most reliable source), we can expect minor influence of forestation on the basin hydrology.

The main issue of this paper is that all the demonstration relies on simulated flows only. Flow simulated over the period 1980-2009 with land-use from 1980 should be compared to actual flow recorded over the period 1980-2009. Another issue is the implicitly presumed stability of the catchment behaviour over each of the 2 periods 1960-79 and 1980-2009. This hypothesis should be further justified. Before modelling, the authors should start their assessment by analyzing actual rainfall and flow data. A graphic showing annual flow, rainfall (both in mm) and runoff coefficients in each of the 3 nested catchments and intermediary catchments (e.g. the colored areas in figure 2) would provide a first assessment of the possible effects of the land-use changes (as done in Lacombe et al. (2008)). A statistical assessment quantifying change and/or trend significance is also missing (cf. Lacombe et al. (2016) for an example).

There is an overall lack of clarity in the writing. The methods used should be explained in more details and with more precision. Figure 1 shows 4 types of treatments for water and soil conservation that occurred in the study area: forestation, terraces, grass and dam. The hydrological impact assessment focuses exclusively on forestation while the 3 others are completely ignored in the analysis. They certainly have altered river flows too. How to account for their effect in the SWAT model? The maps of the study area (figures 2 and 3) do not show where these technics have been implemented.

Splitting the section "Results and discussion" into two distinctive sections "Results" and "Discussion" would certainly help the authors clarifying their scientific demonstration. As it stands, in many places, actual results are juxtaposed with results of previous research which are not referenced.

Detailed comments:

The title should be improved. Currently, it says that LUCC is impacted by the SWAT model.

Abstract: in line 29, it is mentioned that SWAT is applied to the upper and middle reach of the Wei River Basin. It is not clear what is the role of the hydrological station at the outlet of the lower reach.

Introduction

Line 46: a/ the location of the Grain for Green project is missing. b/ Which trees are used for the reforestation? This information is important because, depending on the trees (e.g. deciduous or not), their effect on seasonal flow may be different. c/ the mode of forestation is also primordial when assessing hydrological impacts. For example, natural forest regrowth or tree plantation can have opposite hydrological effects, depending on how the soil is altered. (cf. Lacombe et al. 2016). The authors should provide more details on the type of forestation.

Lines 62-65 do not provide much information, saying that streamflow can increase

whether the vegetation increases or decreases. Too many references here, should be split in two groups (case studies with vegetation increase and case studies with vegetation decrease).

Line 73-75. I don't think that catchment size is the primary control influencing the direction of flow change following land-use change. It is more a question of trade-off between modified infiltration rate and evapotranspiration rate which depends on soil structure, surface properties, depth, slope, vegetation species, etc...

Lines 79-82. The explanation lacks clarity. Again, latitude may indirectly control the hydrological impact of land-use change, but this is certainly not the primary key player.

Line 89: it is not clear if 43% corresponds to the total treated area included in the Wei Basin or if 43% of treated areas corresponds to afforestation.

Line 90: This statement should be supported by a figure showing the time series of actual annual flow (cf. main comments).

Lines 91-92: "streamflow" and "observed annual streamflow". Are you referring to the same variable? Please keep using the same wording when referring to the same variable.

Lines 93-95. Description of geology should be included in the section "study area".

Line 96: "And that drying layer is in great water deficit". Why? Reference required.

Lines 95 to 103: The explanations of the contrasting hydrological behaviours between the "earth-rock mountain landscape" and the Loess Plateau are not clear and not convincing. You did not mention the possible role of slope which is very different between the two types of landscape.

Study Area

Lines 117-118: need to explain what the units provided define exactly.

Line 132: MODIS ?

Line 134: cannot see the six types of LUCC in figure 3.

Lines 136-137: Forest area increased by 0.81% only. It is hardly believable that the hydrological impact quantified later (line 270), (annual average reduction of 94 million m3) was caused by this very minor change.

Line 141: unlike what is written, the soil characteristics are not indicated on the map, (only the names of the soil types are provided).

Line 145: meaning of HRUs ?

Lines 154, 239 and 264-265: avoid "and so on".

Line 160: cf. advices provided in my main comments.

Line 179: need to provide much more information on the input data used to run SWAT.

Lines 185-186: what is an "extraction threshold"?

Line 190: if subdivided into 1 HRU, then it is not subdivided. Please clarify.

Page 11: many parameters and initial values used to calibrate the SWAT model were issued from previous research and experiments (e.g. lines 219: "derived from simulated rainfall experiments", 228: "We have done some research", 230: "Based on the experiments", 234: "were gotten based on experiments"). No references and no explanations are provided. We need more details to understand what has been done.

Lines 237: It is not clear how the authors have accounted for the "management operation of forest" which affect "leaf area index [. . .], plant biomass [. . .], age of trees". Need to provide some explanations here. Which management operations are accounted in the model and how do they affect the variables listed here?

Results and discussions

Line 253: It is not clear if the model efficiencies provided correspond to an average for

each hydrological unit or for the whole basin.

Lines 257, 258: unlike what is written, the trend is not obvious in fig. 6. It would be clearer to redraw the figure at the monthly and annual time steps to visualize possible trends over years.

Line 269: it is not clear what is the 20-year period referred here. Calibration and validation periods are 10 years long and simulation period include 30 years. Further explanations are required. Line 270: there are 3 problems here. 1/ it is not clear in which catchment the hydrological change (annual average reduction of 94 million m3) was assessed, upper or middle ?. b/ this hydrological change should be translated into millimeters of runoff reduction to assess its magnitude and significance. c/ the text indicates that this change is caused by forestation. Indeed, it only reflects the change in the model parameters between the calibration/validation and the simulation periods. But, as already indicated, it does not reflect the actual changes that occurred in the catchment.

Line 273: reference required when referring to previous experiments.

Lines 278-279: "30-year average of the streamflow for forest and agricultural land were taken". Please explain what was done exactly here. Are you referring to the two sets of simulated flow described in lines 263-267? or different hydrological units with agricultural land or forest cover for a given period?

Lines 291-294. This paragraph is about method and should be moved in the appropriate section. It is referring to 3 regions. Which ones? Three different approaches are described to define the LUCC scenarios but the results of each approach are not presented. It seems that figures 8, 10 and 12 only present results for approach 1.

Line 306: the authors indicate that the actual change in forest cover calculated using the land-use maps displayed in figure 3 (0.8% increase) would lead to less than 1% change in streamflow. I agree with this realistic statement but: is it consistent with the

hydrological change quantified in line 270?

Lines 314-325. the authors explain differences in hydrological behaviour of the Loess Plateau and earth-rock mountain, based on other publications, but this paragraph is not linked to the result of the study. The authors need to evidence how these distinctive hydrological behaviours influence their results.

Figures

Fig. 1: Areas under different treatments are expressed in 104 hm2 (i.e. squared hectometers?). This is an atypical unit which is different from the unit used for the study area in the text (104 km2). All areas should be provided in same unit to allow easier comparison. It would be clearer to provide the percentage area so that we anticipate the possible effect of the land treatment on the catchment hydrology.

Fig.2. What is the meaning of all small numbers written on the map of the study area? If they correspond to hydrological units, it is surprising to see numbers in the downstream part which is not included in the study area.

Fig. 6. The scale on the X axis is too big: we cannot see the details in the daily flow variations and in the matching between observed and simulated flow. The figure should be bigger or all panels (calibration and verification should be put in the same column to allow larger size.

Fig. 9: What is the meaning of "corresponding proportional change rate"?

References:

Lacombe G, Cappelaere B, Leduc C. 2008. Hydrological impact of water and soil conservation works in the Merguellil catchment of central Tunisia. Journal of Hydrology. 359: 210-224

Lacombe G, Ribolzi O, de Rouw A, Pierret A, Latsachak K, Silvera N, Pham Dinh R, Orange D, Janeau JL, Soulileuth B, Robain H, Taccoen A, Sengphaathith P, Mouche

[Figure]

E, Sengtaheuanghoung O, Tran Duc T, Valentin C. 2016. Contradictory hydrological impacts of afforestation in the humid tropics evidenced by long-term field monitoring and simulation modelling. Hydrology and Earth System Sciences. 20:2691-2704

---

## Referee Comment (RC2) · Anonymous Referee #2 · 2 Sep 2016

The ideas in this paper are interesting and the results obtained have some implications for land use regulation and water resources management. However, this MS still needs some improvements before publication. The detailed comments are as follows: 1. Could you add the assessment of model performance for use period (1980-2009) except calibration and validation periods? 2. Could you provide the water balance (soil moisture, ET, streamflow, baseflow etc.) for each scenario in a Table? And try to analyze how ET change? 3. Part 2.2, the LUCC data were divided into six types which included forest land and shrub land. As we know, similar to forest land, shrub land is also important for water and soil conservation in (semi)arid area. So, could you make a comparison about stream flow change caused by forest and shrub land change? Could

you show more data and function about check dams, reservoirs, water channels, and water conservancy projects from 1980 to 2009, even for the calibration and validation periods? I understand this is a virtual experimental (or scenario) study, but the results would provide some implications for land use policy, and therefore need carefully check anything related with hydrology cycle. To my knowledge, there are a lot of check dams for agriculture catchments on loess plateau, which might change hydrology (streamflow) as well. If they are not considered in calibration and validation periods, SWAT model may get wrong parameters for different land use types even if the model results (streamflow) is correct. Overall, this is an interesting study, which would provide potential helps for land use policy on loess plateau. The results of this study might suggest that grain for green measures should be different for different eco-regions, i.e., trees may suit some places while grasses suits others.

————————————————————

---

## Author Comment (AC1) · 2 Oct 2016

To: Hydrology and Earth System Sciences (HESS) Subject: Revise the manuscript (#hess-2016-332) The Authors: Wang & Sun The Title: Impact of LUCC on Streamflow Based on the SWAT Model over the Wei River Basin on the Loess Plateau of China Response: The authors appreciate Dr. Lacombe for helpful criticism and 8-pages constructive comments that improved our original manuscript. We have addressed the comments below and have made corrections. The changes being made are marked in red in the manuscript. Response to the main comments: 1: First of all, I am questioning the significance of the hydrological changes that actually occurred in the catchment over the studied period. Although figure 1 indicates that forested areas increased by

about 65.104 hm2 (unusual unit used on the Y axis), which is equivalent to 14% of the upper catchment area, figure 3 inconsistently shows that forested area increased by only 0.81% (line 137) over the same period (1980 to 2005). How to explain this difference? If we rely on figure 2 (which is likely the most reliable source), we can expect minor nfluence of forestation on the basin hydrology. Thank you for pointing this out, which makes us think it through. In fact, these two datasets are from different sources. First of all the number in Fig. 1 is for 6.53% (instead of 14%) as shown in detailed explanation below, which is still higher than 0.81% as pointed by the referee. This is a classic issue in the national survey on soil and water conservation where they only take the revegetation implemented into account and ignore for example possible death of vegetation, which is important in the land use map. One more thing is that the national survey when counting the revegetation areas does not consider the vegetation coverage, which is in fact important in any hydrological modelling including the SWAT modelling. We agree with the referee on that point. With that caution, however, Figure 1 is just for a reference on the development of the soil and water conservation project in China. The data we are relying on are the land use maps. Detailed explanation below: (1) There are some detailed descriptions about Fig.1. The same legends for Fig.1 (a) and (b) brought some confusion, so we revised the legends of Fig. 1. Figure 1 is the developing process of the soil and water conservation measures in the main stream basin of Wei River, including the upper and middle reaches ($4.68\times104$ km2) and the downstream of the main stream ($1.65\times104$ km2). Figure 1 involves about $6.33\times104$ km2. Figure 1 (a) is the area developing of forestation, terraces, grass and dam land separately. The area of forestation was about $57.43\times104$ hm2 during 1980s and it increased to $98.75\times104$ km2 in 2006, which equivalent to 6.53% of the main stream basin of Wei River. And Fig. 1 (b) is the sum area of the forestation, terraces, grass and dam land in upstream, midstream and downstream. And the sum area increased by about $66.15\times104$ hm2 in upstream.

Fig. 1 The development of soil and water conservation measures in the main stream basin of Wei River over last 50 years. (2) Figure 1 is the statistical data of government

based on natural forest before and artificial planting area, which involves all planting of forestation without considering canopy density, surviving or deforestation and so on. The forest of the LUCC data refers to the natural forest and plantation, which canopy density is larger than 30% (Table 3: note âŚą). (3) The forest data of Fig. 1 also includes planting land used as agro-fruit, agro-mulberry, agroforestry and replanting land for trees. While land used for agro-fruit, agro-mulberry, agroforestry is classed as Agricultural land (Table 3: note âŚǎ) in LUCC. (4) There are also some screening conditions for land use types dividing in SWAT model. For hydrological response unit (HRU) analyst, the Dominant Land Use method was used for HRU definition. So the dominant unique combination of land use in the subbasin is used to simulate the HRU. Figure 1 shows the area of grass is smaller than forest's, while it is opposite in LUCC and SWAT model attributed to canopy density and the Dominant method. 2ïijŽ(1) The main issue of this paper is that all the demonstration relies on simulated flows only. Flow simulated over the period 1980-2009 with land-use from 1980 should be compared to actual flow recorded over the period 1980-2009. Thank you for your comments. We add a new Fig. 8 to show the time-series graph of calculated streamflow vs. observed streamflow during 1980-2009 for hydrological stations. We can see the calculated streamflow matched well with the observed values during 1980s. The observed values were measured daily based on the actual LUCC, while the calculated streamflow was got based on LUCC of 1980. So Fig. 8 shows the calibrated SWAT model played well in our study area and the changing LUCC can affect streamflow gradually. The streamflow of typical year, the same year with LUCC, is the results of by LUCC and meteorological conditions. To reduce influence of meteorological condition and isolate the impact of the LUCC on streamflow, 30-year average of the streamflow for forest and agricultural land were taken, respectively. For period of 1980-2009, we just used their measured and long-term daily meteorological data in the study area to drive the validated model for the designed hydrological experiments.

Fig. 8 The time-series graphs of calculated vs. observed streamflow during 1980-2009 for hydrological stations. (2) Another issue is the implicitly presumed stability of the

catchment behaviour over each of the 2 periods 1960-79 and 1980-2009. A graphic showing annual flow, rainfall (both in mm) and runoff coefficients in each of the 3 nested catchments and intermediary catchments (e.g. the colored areas in figure 2) would provide a first assessment of the possible effects of the land-use changes (as done in Lacombe et al. (2008)). A statistical assessment quantifying change and/or trend significance is also missing (cf. Lacombe et al. (2016) for an example). Thank you for your comments. For period of 1980-2009, we just used their measured and long-term daily meteorological data of the study area to drive the validated model. There was only one variable (LUCC or vegetation) to analyze its impacts on streamflow quantitatively. So the soil data, DEM and meteorological data are all same. The figures of annual flow, rainfall and runoff coefficients for 3 regions of Fig. 2 in the study area are added as Fig. 6. The annual average precipitation over the region was calculated using Thiessen polygon method of ArcGIS 9.3, which divided the basin and gave the weight of each meteorological station according to its control area (Fig. 6 (a)). And the regional annual average runoff coefficients were 0.13, 0.27 and 0.16 for figure 6 (a), (b) and (c) in turn.

Fig.6 The time-series of precipitation, annual streamflow and runoff coefficients for the study area (3) There is an overall lack of clarity in the writing. The methods used should be explained in more details and with more precision. Figure 1 shows 4 types of treatments for water and soil conservation that occurred in the study area: forestation, terraces, grass and dam. The hydrological impact assessment focuses exclusively on forestation while the 3 others are completely ignored in the analysis. They certainly have altered river flows too. How to account for their effect in the SWAT model? The maps of the study area (figures 2 and 3) do not show where these technics have been implemented. Splitting the section "Results and discussion" into two distinctive sections "Results" and "Discussion" would certainly help the authors clarifying their scientific demonstration. As it stands, in many places, actual results are juxtaposed with results of previous research which are not referenced. Thank you for your criticism. We have revised the manuscript carefully and add more details to make the writing clarify and avoid possible grammar or syntax error. There were measures of forestation, terrace, grass and dam for soil and water conservation. According to Fig.1, we could see the soil and water conservation measures were mainly implemented in the study area after the 1980s in study area. Hence we choose 1960-1969 and 1970-1979 for the model calibration and validation respectively. For period of 1980-2009, we just used their measured and long-term daily meteorological data in the study area to drive the validated model for the designed hydrological experiments. Measures of soil and water conservation are classified according to LUCC types, which are divided into six types and further 25 subtypes. And the six types included forest, pasture, cropland, water body, residential area and bare. For example, the terrace is treated as Agricultural land with different slope. The impacts of terrace and dam on streamflow are clear. But the impacts of vegetation on streamflow are controversial and complicated and results are different among different basins. We also analyzed the impact of grass on streamflow monthly. The result was similar with forest and its impact on stream was smaller than that. So the forest was selected to analyze in detail. Detailed comments: (1) The title should be improved. Currently, it says that LUCC is impacted by the SWAT model. We changed the title to be "Impact of LUCC on Streamflow Based on the SWAT Model over the Wei River Basin on the Loess Plateau of China" (2)Abstract: in line 29, it is mentioned that SWAT is applied to the upper and middle reach of the Wei River Basin. It is not clear what is the role of the hydrological station at the outlet of the lower reach. Thank you for your suggestion. The Linjiacun, Weijiabu and Xianyang hydrological stations are used in our study (Fig. 2). Linjiacun station locates at the control section of the upstream and Xianyang station is the control station of middle reaches (line 135-136). And Weijiabu station locates between them. The hydrological stations of downstream or the outlet of Wei River were not in our study area. Three regions of different colors in Fig. 2 are divided by 3 hydrological stations of upper and middle reaches. (3) Introduction: Line 46: a/ the location of the Grain for Green project is missing. b/ Which trees are used for the reforestation? This information is important because, depending on the trees (e.g. deciduous or not), their effect on seasonal flow may be different. c/ the mode of forestation is also primordial when

assessing hydrological impacts. For exam-ple, natural forest regrowth or tree planta-tion can have opposite hydrological effects, depending on how the soil is altered. (cf. Lacombe et al. 2016). The authors should provide more details on the type of foresta-tion. Thank you for your suggestion. The Grain for Green project involves most area of China, including 1897 counties of 25 provinces (autonomous regions and municipal-ities), which covers our study area entirely. When the LUCC data are classified and re classified in SWAT model, the tree types are summarized as Range-Brush (RNGB), Forest-Mixed (FRST) and Forest-Deciduous (FRSD). Different types have different hy-drological responses for their leaf, roots and so on. We also analyzed the streamflow generation of the main types of forest (RNGB, FRST and FRSD) in study area further. Results showed that the streamflow yield of FRST and FRSD were about 1.20 and 1.60 times of that of RNGB respectively. In Part 2 and 4.1, the forest included all these types, while for the hydrological experiments (part 4.2 and 4.3) the agricultural land was converted into Forest-Mixed (FRST) only. (4) Lines 62-65 do not provide much information, saying that streamflow can increase whether the vegetation increases or decreases. Too many references here, should be split in two groups (case studies with vegetation increase and case studies with vegetation decrease). Done! Thank you for your comments. (Line 64-68). (5) Line 73-75. I don't think that catchment size is the primary control influencing the direction of flow change following land-use change. It is more a question of trade-off between modified infiltration rate and evapotranspira-tion rate which depends on soil structure, surface properties, depth, slope, vegetation species, etc... Thank you for your suggestion. We agree with the referee on that point. Some of them thought it was probably the large amount of transpiration water played the main function in hydrological process when the watershed was smaller. And some thought that the different impacts of area probably because the forest of larger water-shed could increase precipitation and forest was also conducive to the infiltration of water, which increased the proportion of the underground flow of sreamflow in forest region (Line 77-81). (6) Lines 79-82. The explanation lacks clarity. Again, latitude may indirectly control the hydrological impact of land-use change, but this is certainly

not the primary key player. Thank you for your suggestion. More details are used to explain this. Huang (1982) analyzed Soviet research results found that 48% runoff co-efficients increased, 32% has no change, and 20% decreased with watershed forest increasing. The increased regions were located at high latitude and humid areas. Under this condition, the total evaporation in wooded areas and woodless area are equal. The speculation was that snow may be blown away or to wooded areas from woodless area, which could enhance the coefficient of streamflow but these factors would be weaker over low to middle latitude than that in high latitude.(Line 87-91) (7) Line 89: it is not clear if 43% corresponds to the total treated area included in the Wei Basin or if 43% of treated areas corresponds to afforestation. Thank you for your suggestion. It is "more than 43% of the total treated area was the forestation in the main stream of Wei River basin". (Line 101) (8) Line 90: This statement should be supported by a figure showing the time series of actual annual flow (cf. main comments). Thank you for your suggestion. The figure of actual annual flow has been added (Fig.6). (9) Lines 91-92: "streamflow" and "observed annual streamflow". Are you referring to the same variable? Please keep using the same wording when referring to the same variable. Thank you for your suggestion. Done! (Line 104) (10) Lines 93-95. Description of geology should be included in the section "study area". Thank you for your suggestion. (11) Line 96: "And that drying layer is in great water deficit". Why? Reference required. Thank you for your suggestion. A dried soil layer is generally formed in the soil profile at a particular depth owing to serious soil desiccation in water-limited ecosystems. The residual maximum likelihood analysis demonstrated that land use, rainfall, soil type and slope gradient had a significant impact on dried soil layer thickness, while only land use, rainfall, and soil type influenced the dried soil layer depth of formation significantly. (Line 109-111) References: Wang Y., Shao M., Shao H.: A preliminary investigation of the dynamic characteristics of dried soil layers on the Loess Plateau of China, Journal of Hydrology, 381, 9-17,2010 a. Wang Y., Shao M., Liu Z.: Large-scale spatial variability of dried soil layers and related factors across the entire Loess Plateau of China, Geoderma, 159, 99-108, 2010 b. (12) Lines 95 to 103: The explanations of

the contrasting hydrological behaviours between the "earth-rock mountain landscape" and the Loess Plateau are not clear and not convincing. You did not mention the possible role of slope which is very different between the two types of landscape. Thank you for your suggestion. Slope is one of the impact factors and it is also a significant impact on dried soil layer thickness (Line 109-111, Wang, 2010a). And for Loess Plateau, which also has lots of mountains, its infiltration water flowing into river is related to slope indeed, while the amount is smaller than that generated from earth-rock mountain landscape. Study Area (13) Lines 117-118: need to explain what the units provided define exactly. Thank you for your suggestion. It may be clear if the sentence is revised as "We choose basin of the upper and middle reaches (4.68×104 km2) of the Wei River basin (103.97ãĂĆ∼ 108.75ãĂĆE, 33.69ãĂĆ∼ 36.20ãĂĆN, 13.48×104 km2). ". (Line 132-133) (14) Line 132: MODIS ? Thank you for your suggestion. Done! (Line 147) (15) Line 134: cannot see the six types of LUCC in figure 3. Thank you for your suggestion. There are more details about legend of Fig. 3. Figure 3 is preliminary classification results of the 25 subtypes of LUCC types. And then it is classified to the six types including forest, pasture, cropland, water bodies, residential areas and the bare. The corresponding relations between Fig. 3 and these six types are: âŚă The forest type includes Range-Brush (RNGB), Forest-Mixed (FRST), Forest-Deciduous (FRSD), Pine (PINE) and Forest-Evergreen (FRSE); âŚą The pasture type includes Pasture (PAST), Winter Pasture (WPAS) and Range-Grasses (RNGE); âŚć The cropland means Agricultural Land (AGRL); âŚč Water includes water (WATR) and Wetlands-Mixed (WETL); âŚď The residential areas include area of Residential-High Density (URHD) and Residential-Medium Density (URMD); âŚě The code of bare type is BARE. (Line 151-156) (16) Lines 136-137: Forest area increased by 0.81% only. It is hardly believable that thehydrological impact quantified later (line 270), (annual average reduction of 94 million m3) was caused by this very minor change. Thank you for your suggestion. The annual streamflow of 94 million is equal to 2.0 mm/yr for study area, which is the average result of annual streamflow decreased during 20 years. And the average annual streamflow decreased 0.62 mm for all 30 years (1980-

2009). These results are in ranges of existing research result also. (Line 299) (17) Line 141: unlike what is written, the soil characteristics are not indicated on the map,(only the names of the soil types are provided). Thank you for your suggestion. This map means "the soil data map", which is a vector data including much information and did not just Fig. 4 (a). The detailed soil characteristics can be found from data base we offered. There are 83 types of soil in study area and the types are classed according to soil composition, soil particle size and so on. There are some soil characteristics of HRU 1 in study area for example. Land use: AGRL Soil Name: QSHMT Depth [mm]: 120.00 620.00 1280.00 Bulk Density Moist [g/cc]: 1.33 1.46 1.50 Ave. AW Incl. Rock Frag: 0.19 0.18 0.17 Ksat. (est.) [mm/hr]: 16.58 4.93 3.73 Organic Carbon [weight %]: 2.80 1.00 0.50 Clay [weight %]: 23.00 24.00 25.00 Silt [weight %]: 62.00 60.00 58.00 Sand [weight %]: 15.00 16.00 17.00 Rock Fragments [vol. %]: 0.00 0.00 0.00 Soil Albedo (Moist) : 0.16 0.16 0.16 Erosion K : 0.34 0.40 0.34 Salinity (EC, Form 5) : 0.00 0.00 0.00 (18) Line 145: meaning of HRUs ? Thank you for your suggestion. HRUs are Hydrological response units and the full name has been added. (Line 166) (19) Lines 154, 239 and 264-265: avoid "and so on". Thank you for your suggestion. Done! (20) Line 160: cf. advices provided in my main comments. Thank you for your suggestion. Done! (21) Line 179: need to provide much more information on the input data used to run SWAT. Thank you for your suggestion. The input data refers to data involved in last sentence "It is forced with meteorological data and input with soil properties, topography, land use, and land management practices in the catchment". (Line 198-199) (22) Lines 185-186: what is an "extraction threshold"? Thank you for your suggestion. The extraction threshold area defines the minimum drainage area required to form the origin of a stream (Line 208-209). The user has the ability to set the minimum size of the subbasins.This function plays an important role in determining the detail of the stream network and the size and number of sub-watersheds. (Arcswat interface for SWAT 2009 User's guide, 2010). (23) Line 190: if subdivided into 1 HRU, then it is not subdivided. Please clarify. Thank you for your suggestion. Delineate the watershed into subbasins using Digital Elevation Model (DEM) data and define the
HRUs are key and necessary procedures for SWAT model building. Each watershed is first divided into subbasins and then in hydrologic response units (HRUs) based on the land use and soil distributions. And they have different functions. When a watershed is divided into subbasins, lots of information is loaded into the model from five sections: DEM setup, stream definition, outlet and inlet definition, watershed outlet selection and definition and subbasin parameters. And HRU analysis allows users to load land use and soil layers into the model, evaluate slope characteristics, and determine the land use/soil/lope class combinations and distributions for the delineated watershed and each subbasin. (Arcswat interface for SWAT 2009 User's guide, 2010). (24) Page 11: many parameters and initial values used to calibrate the SWAT model wereissued from previous research and experiments (e.g. lines 219: "derived from simulated rainfall experiments", 228: "We have done some research", 230: "Based on the experiments", 234: "were gotten based on experiments"). No references and no explanations are provided. We need more details to understand what has been done. Thank you for your suggestion. Done! (25) Lines 237: It is not clear how the authors have accounted for the "management operation of forest" which affect "leaf area index [...], plant biomass [...], age of trees". Need to provide some explanations here. Which management operations are accounted in the model and how do they affect the variables listed here? Thank you for your suggestion. SWAT model can simulate 15 different types of management operations. The primary file used to summarize the land and water management practices taking place is the HRU management file (.mgt). This file contains input data for planting, harvest, irrigation applications, nutrient applications, pesticide applications, and tillage operations. In our modeling process, the agricultural land includes operations: planting/ beginning of growing season, auto fertilization initialization, harvest and kill operation. The forest just includes planting/ beginning of growing season. The planting/ beginning of growing season operation initialize the growth of a specific land cover/ plant type in the HRU. For example: âŚă HRU 1 Land use: AGRL Operation Schedule: Operation Schedule: 0.150 1 1 967.69930 0.00 0.00000 0.00 0.00 0.00 0.160 11 1 0.75000 0.00 0.00000 0.00 0.00 1.200 5 0.00000 The first line is the planting/ beginning of growing season operation. The parameters of the first four numbers are HUSC, MGT_OP, PLANT_ID, HEAT UNITS in turn. HUSC is the timing of planting operation, which is the fraction of total base zero heat units at which operation takes place. MGT_OP is operation code. MGT_OP=1 is for plant operation. PLANT_ID is plant/ land cover code from crop.dat. PLANT_ID=1 means that the crop is warm season annual legume. For this crop type, the root depth varies during growing season due to root growth and heat unit theory is used to regulate the growth cycle of plants. HEAT UNITS is the total heat units for cover/plant to reach maturity. Temperature is one of the most important factors governing plant growth. For any plant, a minimum or base temperature must be reached before any growth will take place. Above the base temperature the more rapid the growth rate of the plant. Once the optimum temperature is exceeded the growth rate will begin to slow until a maximum temperature is reached at which growth ceases. The heat unit theory postulates that plants have heat requirements that can be quantified and linked to time to maturity. For example, assume sweet peas are growing with a base temperature of 5 oC. If the mean temperature on a given day is 20 oC, the heat units accumulated on that day are 20-5 =15 heat units. MGT_OP=5 is for harvest and kill operation plant operation. This operation harvests the portion of the plant designated as yield, removes the yield from the HRU and converts the remaining plant biomass to residue on the soil surface. The harvest and kill operation stops plant growth in the HRU. The fraction of biomass specified in the land cover's harvest index is removed from HRU as yield. âŚą HRU 307 Land use: FRST Operation Schedule: 0.150 1 6 50 1043.40000 5.00 1000.00000 0.00 0.00 0.00 The parameters of the first seven numbers are HUSC, MGT_OP, PLANT_ID, CURYR_MAT, HEAT UNITS, LAT_INIT, BIO_ INIT in turn. The HUSC and MGT_OP are the same with AGRL. PLANT_ID=6 means that the crop is perennial which root depth always equal to the maximum allowed for the plant species and soil and plant goes dormant when daylength is less than the threshold daylength. CURYR_MAT is the current age of trees (years). LAT_INIT is the intial leaf area index. This variable is used only for covers/ plants which are transplanted rather than established from seeds.

LAI is the leaf area index of the canopy. The plant canopy can significantly affect in-filtration, surface runoff and evaporation. Canopy storage is the water intercepted by vegetative surface where it is held and made available for evaporation. When precip-itation falls on any given day, the canopy storage is filled before any water is allowed to reach ground. Potential soil water evaporation and plant transpiration are estimated as a function of potential evapotranspiration and LAI. The leaf area index (LAI) for the reference crop is estimated using an equation developed by Allen et al. (1989) to cal-culate LAI as a function of canopy height. For trees, the fraction of potential heat units accumulated for the plant on a given day in the growing season, the fraction of growing season, the number of years for the tree species to reach development. BIO_ INIT is the initial dry weight biomass (kg/ha). This variable is used only for covers/ plants which are transplanted rather than established from seeds. The potential increase in plant biomass on a given day is a function of intercepted energy and the plant's effi-ciency in converting energy to biomass. Energy interception is estimated as a function of solar radiation and the plant's LAI. Results and discussions (26) Line 253: It is not clear if the model efficiencies provided correspond to an average for each hydrological unit or for the whole basin. Thank you for your suggestion. They were corresponding statistic results of Fig. 7 (The time-series graphs of calculated vs. observed values during calibration period and verification period for hydrological stations) for each hy-drological station. (Line 275) (27) Lines 257, 258: unlike what is written, the trend is not obvious in fig. 6. It would beclearer to redraw the figure at the monthly and annual time steps to visualize possible trends over years. Thank you for your suggestion. The monthly time-series graph of calculated vs. observed values during calibration period and verification period for hydrological stations is as follows.

Fig. 1.1 The monthly time-series graphs of calculated vs. observed values during cal-ibration period and verification period for hydrological stations (28) Line 269: it is not clear what is the 20-year period referred here. Calibration and validation periods are 10 years long and simulation period include 30 years. Further explanations are required. Line 270: there are 3 problems here. 1/ it is not clear in which catchment the hydrological change (annual average reduction of 94 million m3) was assessed, upper or middle ?. b/ this hydrological change should be translated into millimeters of runoff reduction to assess its magnitude and significance. c/ the text indicates that this change is caused by forestation. Indeed, it only reflects the change in the model parameters between the calibration/validation and the simulation periods. But, as already indicated, it does not reflect the actual changes that occurred in the catchment. Thank you for your suggestion. We have revised this part (Line 298-299). It is 20 years in 30 simulation years, which annual streamflow decreased. In other 10 years, the streamflow did not decrease. (a) It changed in the study area (upper and middle reaches of the Wei River basin). (b) The annual average reduction was 2.0 mm/yr for these years in study area (Li). (c) The text indicates that the change is caused by LUCC and hydrological conditions. Because the LUCC involves too many types of land uses, we then designed the experiments for forest changing only to study its impact. Because under the same hydrological condition, the streamflow reduced in most years and increased in other years, 30-year average of the streamflow for forest and agricultural land were taken, respectively to reduce influence of meteorological conditions and isolate the impact of the LUCC on streamflow. (29) Line 273: reference required when referring to previous experiments. Thank you for your suggestion. Done! (30) Lines 278-279: "30-year average of the streamflow for forest and agricultural land weretaken". Please explain what was done exactly here. Are you referring to the two sets of simulated flow described in lines 263-267? or different hydrological units with agricultural land or forest cover for a given period? Thank you for your suggestion. The 30-year (1980-2009) average values of the streamflow for forest and agricultural land were averaged respectively. (Line 308-309) (31) Lines 291-294. This paragraph is about method and should be moved in the appropriate section. It is referring to 3 regions. Which ones? Three different approaches re described to define the LUCC scenarios but the results of each approach are not resented. It seems that figures 8, 10 and 12 only present results for approach 1. Thank you for your suggestion. The 3 regions were divided by 3 Linjiacun, Weijiabu and Xianyang hydrological stations (three different color regions with number)

(Line 331-332). They were 3 control conditions when the land use converted from agricultural land to forest. The second and third conditions were considered as much as possible to reduce impacts of other factors. (32) Line 306: the authors indicate that the actual change in forest cover calculated using he land-use maps displayed in figure 3 (0.8% increase) would lead to less than 1% hange in streamflow. I agree with this realistic statement but: is it consistent with the hydrological change quantified in line 270? Thank you for your suggestion. Line 306 is the result of conversion of agricultural land to forest on streamflow. Line 270 is the result of LUCC changes on streamflow, which involves many types of land use conversion measures and is a balanced result among these measures. So the changes of streamflow, surface runoff, soil flow and baseflow between agricultural land and forest were singled out (Fig. 9 The changes of 30-year (1980-2009) averages of streamflow, surface runoff, soil flow and baseflow between agricultural land and forest.). We can see the impacts are consistent. (33) Lines 314-325. the authors explain differences in hydrological behaviour of the Loess Plateau and earth-rock mountain, based on other publications, but this paragraph is not linked to the result of the study. The authors need to evidence how these distinctive hydrological behaviours influence their results. Thank you for your suggestion. As suggested, we revised the whole manuscript carefully and add more details to make it clean. Figures (34) Fig. 1: Areas under different treatments are expressed in 104 hm2 (i.e. squared hectometers?). This is an atypical unit which is different from the unit used for the study area in the text (104 km2). All areas should be provided in same unit to allow easier comparison. It would be clearer to provide the percentage area so that we anticipate the possible effect of the land treatment on the catchment hydrology. Thank you for your suggestion. Figure 1 is revised as suggestion.

Fig.1 (35) Fig.2. What is the meaning of all small numbers written on the map of the study area? If they correspond to hydrological units, it is surprising to see numbers in the downstream part which is not included in the study area. Thank you for your suggestion. The numbers of Fig. 2 are serial number of subbasins/ HRUs (Line 208). All numbered area is study area. Linjiacun station locates at the control section of

the upstream and Xianyang station is the control station of middle reaches (Line 133-135). And the upper and middle reaches of the Wei River basin is the study area. (36) Fig. 6. The scale on the X axis is too big: we cannot see the details in the daily flow variations and in the matching between observed and simulated flow. The figure should be bigger or all panels (calibration and verification should be put in the same column to allow larger size. Thank you for your suggestion. It is more clearly indeed as suggestion (Fig. 7).

Fig. 7 (37) Fig. 9: What is the meaning of "corresponding proportional change rate"? Thank you for your suggestion. It is the change rate of streamflow at the Linjiacun, Weijiabu and Xianyang stations correspondingly. We have revised the figure (Fig.11).

References: Lacombe G, Cappelaere B, Leduc C. 2008. Hydrological impact of water and soil conservation works in the Merguellil catchment of central Tunisia. Journal of Hydrology. 359: 210-224. Lacombe G, Ribolzi O, de Rouw A, Pierret A, Latsachak K, Silvera N, Pham Dinh R, Orange D, Janeau JL, Soulileuth B, Robain H, Taccoen A, Sengphaathith P, Mouche E, Sengtaheuanghoung O, Tran Duc T, Valentin C. 2016. Contradictory hydrological impacts of afforestation in the humid tropics evidenced by long-term field monitoring and simulation modelling. Hydrology and Earth System Sciences. 20:2691-2704. Thank you for your recommendations. The references have been cited (Line 53-54, line 84)..
* * *
[Figure]

[Figure]

Fig. 1 The development of soil and water conservation measures in the main stream basin of Wei River over last 50 years.

**Fig. 1.**

---

## Author Comment (AC2) · 2 Oct 2016

To: Hydrology and Earth System Sciences (HESS) Subject: Revise the manuscript (#hess-2016-332) The Authors: Wang& Sun The Title: Impact of LUCC on Streamflow Based on the SWAT Model over the Wei River Basin on the Loess Plateau of China Response: The authors appreciate the reviewers for helpful and constructive comments that improved our original manuscript. We have addressed the comments below and have made corrections. The changes being made are marked in red in the manuscript. Response to the detailed comments: 1. Could you add the assessment of model performance for use period (1980-2009) except calibration and validation periods? Thank you for your suggestions. We add a new Fig. 8 to show the time-series graph of calculated streamflow vs. observed streamflow during 1980-2009 for hydrological stations. We can see the calculated streamflow matched well with the observed values before 1990. The observed values were measured daily based on the actual LUCC, while the calculated streamflow was got based on LUCC of 1980. So Fig. 8 shows the calibrated SWAT model played well in our study area and the changing LUCC can affect streamflow gradually. The streamflow of typical year, the same year with LUCC, is the results of by LUCC and meteorological conditions. To reduce influence of meteorological condition and isolate the impact of the LUCC on streamflow, 30-year average of the streamflow for forest and agricultural land were taken, respectively. For period of 1980-2009, we just used their measured and long-term daily meteorological data in the study area to drive the validated model for the designed hydrological experiments.

Fig.8 The time-series graphs of calculated vs. observed streamflow during 1980-2009 for hydrological stations. 2. Could you provide the water balance (soil moisture, ET, streamflow, baseflow etc.) for each scenario in a Table? And try to analyze how ET change? Thank you for your suggestions. Table 2.1 shows the water balance for different scenarios. The ET values decreased with with increasing of forest area overall. Table 2.1 The water balance for different scenarios S1 S2 S3 S4 S5 ET (mm) 388.98 380.39 373.38 358.87 311.47 Surface runoff (mm) 21.19 21.13 21.43 21.58 21.53 Soil flow (mm) 68.42 69.52 70.63 72.57 77.22 Baseflow (mm) 29.92 36.99 42.37 54.06 94.24 Precipitation (mm) 509.62

3. Part 2.2, the LUCC data were divided into six types which included forest land and shrub land. As we know, similar to forest land, shrub land is also important for water and soil conservation in (semi)arid area. So, could you make a comparison about stream flow change caused by forest and shrub land change? Could you show more data and function about check dams, reservoirs, water channels, and water conservancy projects from 1980 to 2009, even for the calibration and validation periods? I understand this is a virtual experimental (or scenario) study, but the results would provide some implications for land use policy, and therefore need carefully check anyInteractive comment

thing related with hydrology cycle. To my knowledge, there are a lot of check dams for agriculture catchments on loess plateau, which might change hydrology (stream-flow) as well. If they are not considered in calibration and validation periods, SWAT model may get wrong parameters for different land use types even if the model results (streamflow) is correct. Thank you for your suggestions. The forest type includes Range-Brush (RNGB), Forest-Mixed (FRST), Forest-Deciduous (FRSD), Pine (PINE) and Forest-Evergreen (FRSE). In Part 2 and 4.1, the forest included all these types, while for the hydrological experiments (part 4.2 and 4.3) the agricultural land was converted to FRST only. The comparison of per unit streamflow between forest and shrub land for 2 LUCC types from 1980 to 2009 is showed in box figure as figure 2.1. The annual average streamflow increased 0.81% in Range-Brush (RNGB) land and the streamflow yield of forest is about 1.18 times of that of RNGB respectively. We also analyzed the streamflow generation of the main types of forest (RNGB, FRST and FRSD) in study area further. Results showed that the streamflow yield of FRST and FRSD were about 1.20 and 1.60 times of that of RNGB respectively.

Figure 2.1 The per unit streamflow generation between forest and shrub land for 2 LUCC types Figure 2.2 showed the development of different soil and water conservation measures (including forestation, terraces, grass and dam land) in the whole and main stream basin of Wei River respectively. According to this figure, we could see the soil and water conservation measures were mainly implemented in the study area after the 1980s in study area. Hence we choose 1960-1969 and 1970-1979 for the model calibration and validation respectively. For period of 1980-2009, we just used their measured and long-term daily meteorological data in the study area to drive the validated model for the designed hydrological experiments. The long-term data could reduce influence caused by meteorological conditions and isolate the impact of the LUCC on streamflow. Figure 1 is the statistical data of government based on natural forest before and artificial planting, which involves all planting area of forestation and does not consider canopy density. The forest of the LUCC data refers to the natural forest and plantation, which canopy density is larger than 30% (Table 3: note ⌹). Data
of Fig. 1 also includes planting land used as agro-fruit, agro-mulberry, agroforestry and replanting land for trees without surviving or deforestation and so on. But land used for agro-fruit, agro-mulberry, agroforestry is classed as Agricultural land (Table 3: note âŚă). There is also screening condition in SWAT model. For hydrological response unit (HRU) analyst, the Dominant Land Use method was used for HRU definition. So the dominant unique combination of land use in the subbasin is used to simulate the HRU.

Figure 2.2 The development of different soil and water conservation measures in the whole and main stream basin of Wei River respectively.
Interactive

comment

Fig.8 The time-series graphs of calculated vs. observed streamflow during 1980-2009 for hydrological stations.

Table 2.1 The water balance for different scenarios

|                    |                     | S1     | S2     | S3     | S 4 | S5     |
|--------------------|---------------------|--------|--------|--------|------------|--------|
| ET (mm)            |                     | 388.98 | 380.39 | 373.38 | 358.87     | 311.47 |
|                    | Surface runoff (mm) | 21.19  | 21.13  | 21.43  | 21.58      | 21.53  |
| Streamflow         | Soil flow (mm)      | 68.42  | 69.52  | 70.63  | 72.57      | 77.22  |
| (mm)               | Baseflow (mm)       | 29.92  | 36.99  | 42.37  | 54.06      | 94.24  |
| Precipitation (mm) |                     | 509.62 |        |        |            |        |

Fig. 1.

---

## Author Response (AR1)

**To: Hydrology and Earth System Sciences (HESS)**

**Subject:** Revise the manuscript (#hess-2016-332)

The Authors: Wang & Sun

The Title: Impact of LUCC on Streamflow Based on the SWAT Model over the Wei River Basin on the Loess Plateau of China

**Response:**

The authors appreciate the editor and reviews for helpful criticism and constructive comments that improved our original manuscript. We have addressed the comments below (part 1) and supplemented details for review #1 (part 2: Author's Response-RC1) and review #2 (part 3: Author's Response-RC2). And the changes being made are marked in the manuscript (part 4: a marked-up manuscript).

**Part 1**

**Response to the comments:**

1. In your response to reviewer #1, I don't think it is appropriate to use the Thiessen polygon method to calculate basin-scale precipitation. The SWAT allocate each subbasin with a precipitation gauge (or using the lapse rate and elevation bands), therefore, using the SWAT generated precipitation will be more appropriate.

Thank you for this criticism and this is an important suggestion for keeping consistency and continuity though the manuscript. We have recalculated the average values of regional precipitation using elevation bands method of ArcSWAT 2009.93.7b, which can account for orographic effects on precipitation (Neitsch et al., 2011). (Figure 6 and Line 184-204 in marked-up manuscript).

Orographic precipitation is a significant phenomenon in certain areas of the world. To account for orographic effects on both precipitation and temperature, SWAT allows the subbasin to be split into a maximum of ten elevation bands. Precipitation and maximum and minimum temperatures are calculated for each band as a function of the respective lapse rate and the difference between the gage elevation and the average elevation specified for the band.

$$R_{band} = R_{day} + (EL_{band} - EL_{gage}) \cdot \frac{plaps}{days_{pcp,yr} \cdot 1000} \text{ when } R_{day} > 0.01$$

where  $R_{band}$  is the precipitation falling in the elevation band (mm H2O),  $R_{day}$  is the precipitation recorded at the gage or generated from gage data (mm H2O),  $EL_{band}$  is the mean elevation in the elevation band (m),  $EL_{gage}$  is the elevation at the recoeding gage (m), plaps is the precipitation lapse rata (mm H2O/km),  $days_{pcp,yr}$  is the average number of days of precipitation in the subbasin in a year, and 1000 is a factor needed to convert meters to kilometers.

Once the precipitation values have been calculated for each elevation band in the subbasin, new average subbasin precipitation value is calculated:

$$R_{day} = \sum_{bnd=1}^{b} R_{band} \cdot fr_{bnd}$$

where  $R_{day}$  is the daily average precipitation adjusted for orographic effects (mm, H2O),  $fr_{bnd}$  is the fraction of subbasin area within the elevation band, and b is the total number of elevation bands in the subbasin.

**Reference:**

Neitsch, S. L., Arnold, J. G., Kiniry, J. R., Williams, J. R.: Soil and Water Assessment Tool (SWAT) Theoretical Documentation: Version 2009, Texas Water Resources Institute Technical Report No. 406, 2011.

**2. "The impacts of terrace and dam on streamflow are clear". Please clarify and add convincing reference.**

Thank you for your suggestion. We have added more details and references (Line 105-108 in marked-up manuscript). The impacts of terrace and check dam had a regular and negative effect on annual streamflow. They could reduce the runoff in the flood season, increased the baseflow and guarantee the river ecological flows in non-flood season on the Loess Plateau (*Shao, et al., 2012, 2013a, 2013b; Zhang, et al., 2014a, 2014b; Xu, et al., 2013*). For example:

**(1) The impact of terrace on streamflow:**

In order to address the lack of tools in researching terrace impact on watershed soil and water loss, a process-based terrace algorithm within the SWAT model was developed by Shao Baffaut and Gao et al (2012,2013a), which has been incorporated into SWAT (version 2009). The responses of soil and water loss toward terraces over the Weihe River basin were detected using this verified model (*Shao, 2013b*). Results showed the terrace in the main Weihe River basin could delay the flood and add the drought season runoff, which reduced the annual streamflow in

general. Terrace in 2000 could decrease about 37 million  $m^3$  annual water yield in the whole watershed and increased the most dry month runoff by 3.5% in the Xianyang station. Zhang et al (2014a, 2014b) used this model to study the terrace measures of Yanhe river watershed, typical basin of the Loess Plateau, and results showed that the terrace measures could reduce the runoff in the flood season, increased the base flow and guarantee the river ecological flows in non-flood season. And the 1 m3 water could be supplied to the river while 5~ 6 m3 water stored by the terrace of Yanhe river watershed.

References:

Shao, H., Baffaut, C., Gao, J. E.: A Process-Based Method for Evaluating Terrace Runoff and Sediment Yield [J], 2012.

Shao, H., Baffaut, C., Gao, J. E., et al. Development and application of algorithms for simulating terraces within SWAT [J]. Transactions of the Asabe, 2013a, 56(5):1715-1730.

Shao, H.: Simulation of Soil and Water Loss Variation toward Terrace Practice in the Weihe River Basin, Doctor, Northwest A & F University, Yangling Shaanxi, 2013b.

Zhang, Y. X., Gao, J. E., Shao, H., et al. The Terraced Fields Environmental Impact Assessment in Data-Scarce Areas Based on the Embedded Terraced Module SWAT Model [J]. Nature Environment & Pollution Technology, 2014a.

Zhang, Y. X.: The research of watershed runoff and sediments variation toward to the soil and water conservation terrace measure, Doctor, Northwest A & F University, Yangling Shaanxi, 2014b.

**(2) The impact of check dam on streamflow:**

Xu and Fu et al (2013) applied the SWAT (Soil and Water Assessment Tool) model to simulate the streamflow in the Yanhe watershed and results showed that the check dams had a regulation effect on streamflow. From 1984 to 1987, the streamflow in rainy season (from May to October) decreased by  $1.54 \text{ m}^3 \text{s}^{-1}$  (14.7 %) to  $3.13 \text{ m}^3 \text{s}^{-1}$  (25.9 %) due to the check dams; while in dry season (from November to the following April), streamflow increased by  $1.46 \text{ m}^3 \text{s}^{-1}$  (60.5%) to  $1.95 \text{ m}^3 \text{s}^{-1}$  (101.2 %); From 2006 to 2008, the streamflow in rainy season decreased by  $0.79 \text{ m}^3 \text{s}^{-1}$  (15.5 %) to  $1.75 \text{ m}^3 \text{s}^{-1}$  (28.9 %), and the streamflow in dry season increased by  $0.51 \text{ m}^3 \text{s}^{-1}$  (20.1 %) to  $0.97 \text{ m}^3 \text{s}^{-1}$  (46.4 %).

References:

Xu, Y. D., Fu, B. J., He, C. S.: Assessing the hydrological effect of the check dams in the Loess Plateau, China by model simulations, Hydrology & Earth System Sciences Discussions, 2013, 9(12):13491-13517.

But the impacts of vegetation on streamflow are controversial and complicated and results are different among different basins. So the forest was selected to analyze in detail.

3. Some items have been left out. For example, "Splitting the section "Results and discussion" into two distinctive sections "Results" and "Discussion" would certainly help the authors clarifying their scientific demonstration" In reviewer #1; "more information on check dams" in review #2. Please provide a point to point response to these comments.

Thank you for your suggestion. We have revised the manuscript completely and provide a point to point response to the comments (in Author's Response-RC1 and Author's Response-RC2). And the changes being made are marked in the manuscript.

**4. Please clarify what's the dominate vegetation for Range-Brush, Forest-mixed, forest-deciduous, etc.**

Thank you for your suggestion. There are 79 plant species in the plant growth database of SWAT 2009 version. The generic land covers in the model are: RNGB uses values for Little Bluestem (LAImax = 2.0); FRST and FRSD use values for oak; FRSE uses values for pine (Arnold et al., 2011). And the specific and dominate vegetation for Range-Brush, Forest-mixed, forest-deciduous, etc. in study area are as follows:

- Range-Brush (RNGB): Vitex negundo L. var. heterophylla (Franch.) Rehd, Exochorda racemosa (Lindl.) Rehd, Forsythia suspense, Quercus variabilis Bl., Platycarya strobilacea Sieb.et Zucc., Lespedeza Formosa, Abelia parvifolia, Corylus mandshurica, Lindera obtusiloba, etc.
- (2) Forest-Mixed (FRST): Oak, Quercus aliena var. acuteserrata, Platycarya strobilacea Sieb. et Zucc., Pinus armandi, etc.
- (3) Forest-Deciduous (FRSD): Larix gmelinii (Rupr.) Kuzen, Acer Davidii, Juglans cathayensis Dode, Morus alba L., Toxicodendron vernicifluum, etc.
- (4) Pine (PINE): Pinus tabuliformis Carrière, Larix principis-rupprechtii, Larix principis-rupprechtii, etc.
- (5) Forest-Evergreen (FRSE): Abies fabri (Mast.) Craib, Picea asperata Mast., etc.

(Note: The Vegetation type data is provided by Data Center for Resources and Environmental Sciences, Chinese Academy of Sciences (RESDC))

5. A lot of unreadable characters are found in your response. Please do correct them.

Thank you for your suggestion. We have checked and revised them.

6. Check "the annual streamflow is 2.0 mm/yr for study area". It seems too little for the WeiRiver.

Thank you for your suggestion. We have checked the result. The annual average reduction of streamflow was 94 million  $m^3$  in study area and the area of the study basin is  $4.68 \times 10^4 \text{ km}^2$ .

$$\frac{94 \times 10^6 \text{ m}^3}{4.68 \times 10^4 \times 10^6 \text{ m}^2} \times 10^3 \frac{\text{m}}{\text{mm}} = 2.0085 \text{ mm}$$

So it was about 2.0 mm/yr to in study area.

**7. One figure showing the observed and simulated runoff/streamflow is enough.**

Thank you. As your suggestion, only one figure showing the observed and simulated streamflow was kept in the manuscript.

**Part 2: Author's Response-RC1**

**Response:**

The authors appreciate Dr. Lacombe for helpful criticism and 8-pages constructive comments that improved our original manuscript. We have addressed the comments below and have made corrections. The changes being made are marked in red in the manuscript.

**Response to the main comments:**

1: First of all, I am questioning the significance of the hydrological changes that actually occurred in the catchment over the studied period. Although figure 1 indicates that forested areas increased by about 65.104 hm2 (unusual unit used on the Y axis), which is equivalent to 14% of the upper catchment area, figure 3 inconsistently shows that forested area increased by only 0.81% (line 137) over the same period (1980 to 2005). How to explain this difference? If we rely on figure 2 (which is likely the most reliable source), we can expect minor nfluence of forestation on the basin hydrology.

Thank you for pointing this out, which makes us think it through. In fact, these two datasets are from different sources. First of all the number in Fig. 1 is for 6.53% (instead of 14%) as shown in detailed explanation below, which is still higher than 0.81% as pointed by the referee. This is a classic issue in the national survey on soil and water conservation where they only take the revegetation implemented into account and ignores for example possible death of vegetation, which is important in the land use map. One more thing is that the national survey when counting the revegetation areas does not consider the vegetation coverage, which is in fact important in any hydrological modelling including the SWAT model. We agree with the referee on that point. With that caution, however, Figure 1 is just for a reference on the development of the soil and water conservation project in China. The data we are relying on are the land use maps. Detailed explanation below:

(1) There are some detailed descriptions about Fig.1. The same legends for Fig.1 (a) and (b) brought some confusion, so we revised the legends of Fig. 1. Figure 1 is the developing process of the soil and water conservation measures in the main stream basin of Wei River, including the upper and middle reaches  $(4.68 \times 10^4 \text{ km}^2)$  and the downstream of the main stream  $(1.65 \times 10^4 \text{ km}^2)$ . Figure 1 involves about  $6.33 \times 10^4 \text{ km}^2$ . Figure 1 (a) is the area developing of forestation, terraces,

grass and dam land separately. The area of forestation was about  $57.43 \times 10^4$  hm2 during 1980s and it increased to  $98.75 \times 10^4$  km2 in 2006, which equivalent to 6.53% of the main stream basin of Wei River. And Fig. 1 (b) is the sum area of the forestation, terraces, grass and dam land in upstream, midstream and downstream. And the sum area increased by about  $66.15 \times 10^4$  hm2 in upstream.

Fig. 1 The development of soil and water conservation measures in the main stream basin of Wei River over last 50 years

(2) Figure 1 is the statistical data of government based on natural forest before and artificial planting area, which involves all planting of forestation without considering canopy density, surviving or deforestation and so on. The forest of the LUCC data refers to the natural forest and plantation, which canopy density is larger than 30% (Table 3: note 2).

(3) The forest data of Fig. 1 also includes planting land used as agro-fruit, agro-mulberry, agroforestry and replanting land for trees. While land used for agro-fruit, agro-mulberry, agroforestry is classed as Agricultural land (Table 3: note ①) in LUCC.

(4) There are also some screening conditions for land use types dividing in SWAT model. For hydrological response unit (HRU) analyst, the Dominant Land Use method was used for HRU definition. So the dominant unique combination of land use in the subbasin is used to simulate the HRU. Figure 1 shows the area of grass is smaller than forest's, while it is opposite in LUCC and SWAT model attributed to canopy density and the dominant method.

2: (1) The main issue of this paper is that all the demonstration relies on simulated flows only. Flow simulated over the period 1980-2009 with land-use from 1980 should be compared to actual flow recorded over the period 1980-2009.

Thank you for your comments. We add a new figure (Fig. 1.1) to show the time-series graph

of calculated streamflow vs. observed streamflow during 1980-2009 for hydrological stations. We can see the calculated streamflow matched well with the observed values during 1980s. The observed values were measured daily based on the actual LUCC, while the calculated streamflow was got based on LUCC of 1980. So Fig. 1.1 shows the calibrated SWAT model played well in our study area and the changing LUCC can affect streamflow gradually. The streamflow of typical year, the same year with LUCC, is the results of by LUCC and meteorological conditions. To reduce influence of meteorological condition and isolate the impact of the LUCC on streamflow, 30-year average of the streamflow for forest and agricultural land were taken, respectively. For period of 1980-2009, we just used their measured and long-term daily meteorological data in the study area to drive the validated model for the designed hydrological experiments.

---

## Referee Report (RR1)

Reviewer Comments:
Thanks to the editor for giving opportunity to review this article which is already peered first round.

The article entitled: Impact of LUCC on Streamflow Based on the SWAT Model over the Wei River Basin on the Loess Plateau of China by Hong Wang and Fubao Sun. This article is narrated very well and easy to understand. Since it is already reviewed earlier the quality of the paper improved a lot compare to initial submission. Paper is worth of publishing in HESS with minor revision as follows:

1. Sufficient details related to parameterization of SWAT model for different years should be included in the Model Setup. Like how catchment characteristics were obtained for different years for parameterization of the SWAT.
2. How catchment characteristics like land use land cover, slope soil, drainage etc., have been adopted for analysis period i.e from 1980-2009, as these are very important for any hydrological phenomenon in addition to the meteorological parameters.
3. Simulated stream flow results show a high coefficient of variation. Therefore, extent of uncertainty in simulated stream flow results may also be given before commenting on usefulness of study.
4. China's Grain for Green project, represents the anthropogenic changes in hydrology, its impacts should be discussed and results of these impacts leading to conflict results.
5. Latest article by J. Yin et al.2017 in HESS: Effects of land use/land cover and climate changes also shows the similar study results, this paper may also be referred here and results may also compared in discussion, which strengthens the quality of this article.

In line 31: Model was calibrated for 10 years form 1960-69, validated for 1970-79 and even analysis is also a part of validation. Hence it is suggested to validate up to 2009. Large data used from1960-09 (50yrs).

In line 37: What is soil flow, how it differs from base flow? Explain. In my view baseflow and soil flow were one and the same.

Discussion: Reforestation reduced the surface runoff because of vegetation cover and increased streamflow may be because of increase in baseflow that depends on land topography, forestation might be on hillocks. The impact of Grain for Green project shows the conflict results i.e initially from 1969-1980 streamflow reduced and for present periods increasing. And also the Mountain rock regions, higher altitude variation including changing rainfall patter with intensive storms too leading to higher stream flow.

Suggestions: One hydrological cycle represents for 30 years. In this study 50 years data used which too large. Study may be restricted to latest 30 years i.e 1980-2009. Calibration and validation may be carried out for the latest data as it represents' the real conditions specially rainfall distribution pattern.

Finally this article, Impact of LUCC on Streamflow Based on the SWAT Model over the Wei River Basin on the Loess Plateau of China by Hong Wang and Fubao Sun is worth of publishing in HESS with minor revisions as suggested.

---

## Author Response (AR2)

**To:** *Hydrology and Earth System Sciences (HESS)*

**Subject:** Revise the manuscript (#hess-2016-332)

**The Authors:** Wang & Sun

**The Title:** Impact of LUCC on Streamflow Based on the SWAT Model over the Wei River Basin on the Loess Plateau of China

**Dear Prof. Chen and Referees,**

   Thank you for your useful suggestions and constructive comments on our manuscript. We have modified the manuscript accordingly, and the detailed corrections are listed below point by point. Part 1 is a point-by-point response to the reviews and part 2 is a marked-up manuscript version.

**Part 1. A point-by-point response to the reviews**

**Comments to the Author:**

*I think the authors addressed most questions from the two reviewers except the following one from reviewer #1:*
*Another issue is the implicitly presumed stability of the catchment behavior over each of the 2 periods 1960-79 and 1980-2009. A graphic showing annual flow, rainfall (both in mm) and runoff coefficients in each of the 3 nested catchments and intermediary catchments (e.g. the colored areas in figure 2) would provide a first assessment of the possible effects of the land-use changes (as done in). A statistical assessment quantifying change and/or trend significance is also missing (cf. Lacombe et al. (2016) for an example).*

   Thank you for your comments. We have revised the figure (Fig. 5) and statistical assessment according to references (*Lacombe et al., 2008 and 2010*) in part "*2.4 Meteorological and hydrological data*" (line 182-187 and 193-199) and *part "4.1 Impact of the observed LUCC on streamflow"* (line 304-307 and 318-320).

   The supplemented details are as follow:

   Figure 6 (b), (c) and (d) show the time-series of average precipitation calculated though elevation bands method of ArcSWAT from 1960 to 2009. The average of precipitation of region 1, 2 and 3 were 489.71 493.25 and 566.60 mm/yr and the trend analysis showed that the precipitation of them decreased with an average decreasing rate of 0.57, 0.55 and 0.21 mm/yr, whereas the decreasing tendencies were not significant at the 0.05 level.

   The trend analysis showed that streamflow of region 1 and 2 decreased extremely significantly (P < 0.01), with an average decreasing rate of 1.74 and 5.38 mm/yr. The streamflow of region 3 did not decreased significantly. And the average runoff coefficients were 0.13, 0.34 and 0.17 in region 1, 2 and 3 over the past 50 years (1960-2009). The trend analysis of runoff coefficients showed that the tendencies of region 1 and 2 decreased extremely significantly (P < 0.01), with an average decreasing rate of 0.34%, and 1.09 % per year. The runoff coefficient of region 3 decreased significantly (P < 0.01) too, with an average decreasing rate of 0.2% per year.

Analysis above (Fig.6) showed that the observed precipitation of study area did not decreased significantly from 1960 to 2009, while the annual streamflow (except region 3) and runoff coefficients decreased significantly (P < 0.05) under this meteorological conation. This discrepancy could attribute to LUCC changes mostly.

This result was consistent with the decreasing tendencies of the observed streamflow of Xianyang station, which decreased significantly (P < 0.05), with an average decreasing rate of 2.45 mm/yr from 1980 to 2009.

[Figure]

Figure 5 The time-series of precipitation, annual streamflow and runoff coefficients for the regions of study area

*In addition, the combined effects of forestation, terraces, grass and dam land on hydrological*

*processes should be discussed.*

Thank you for your suggestion. We have added the combined effects of forestation, terraces, grass and dam land on hydrological processes in part "*4.2 Hydrological experiments on the impact of conversion of agricultural land to forests on streamflow*" (Line 411-433).

The supplemented details are as follow:

Seemingly, this result was not consistent with the significant decreasing tendencies of streamflow in study area. The combined effects of LUCC, including forestation, terraces, grass, and dam, could explain the discrepancy. Under the same meteorological conditions, the streamflow is mainly a result of combined effects of these measures. Results showed the terrace in the main Weihe River basin could delay the flood and add the drought season streamflow, which reduced the annual streamflow in general. The terrace in 2000 could decrease about 37 million $m^3$ annual water and increased the most dry month streamflow by 3.5% in Xianyang station (*Shao, 2013b*). Zhang et al (*2014a, 2014b*) studied the terrace measures of Yanhe River basin, typical basin of the Loess Plateau, and results showed that the terrace measures could reduce the runoff in the flood season and increased the baseflow. Results showed that 1 $m^3$ water could be supplied to the river when 5~ 6 $m^3$ water stored by the terrace. This meant the water reducing effect of terrace was larger than 80% in Yanhe River basin and. Xu et al. (2012) applied the SWAT model to simulate the streamflow in the Yanhe basin and results showed that the check dams had a regulation effect on streamflow. From 1984 to 1987, the streamflow in rainy season (from May to October) decreased by 1.54 $m^3s^{-1}$ (14.7 %) to 3.13 $m^3s^{-1}$ (25.9 %) due to the check dams; while in dry season (from November to the following April), streamflow increased by 1.46 $m^3s^{-1}$ (60.5%) to 1.95 $m^3s^{-1}$ (101.2 %); From 2006 to 2008, the streamflow in rainy season decreased by 0.79 $m^3s^{-1}$ (15.5 %) to 1.75 $m^3s^{-1}$ (28.9 %), and the streamflow in dry season increased by 0.51 $m^3s^{-1}$ (20.1 %) to 0.97 $m^3s^{-1}$ (46.4 %). Lots of results showed that the terrace and check dam both had a negative effect on annual streamflow which was a result of the balance between the streamflow reducing in the flood season and baseflow increasing in non-flood season on the Loess Plateau (Shao, et al., 2012, 2013a, 2013b; Zhang, et al., 2014a, 2014b; Xu, et al., 2012). The observed streamflow was a result of the balance among forestation, terraces, grass, and dam.

**Specific comments:**

*1. Figure 1: the subbasin numbers are not necessary.*

We have revised the Figure 1 as suggestion. Thank you.

*2. Provide the statistical assessments of the model performance besides Figure 6.*

Thank you for your suggestion. The statistical assessments have been supplemented in the test (Line 183-187 and 193-199).

**Author's Response- reviewer #1**
*The article entitled: Impact of LUCC on Streamflow Based on the SWAT Model over the Wei River Basin on the Loess Plateau of China by Hong Wang and Fubao Sun. This article is narrated very well and easy to understand. Since it is already reviewed earlier the quality of the paper improved a lot compare to initial submission. Paper is worth of publishing in HESS with minor revision as follows:*
*1. Sufficient details related to parameterization of SWAT model for different years should be included in the Model Setup. Like how catchment characteristics were obtained for different years for parameterization of the SWAT.*

We are sorry for insufficient information to bother you. For the SWAT model, the parameters did not change for different years, while they were changed with different slopes, soli type and LUCC in different HRU. Linjiacun, Weijiabu and Xianyang hydrological stations were loaded manually as subbasin outlets in our test. The comprehensive parameters which were different in different slopes, soli type and LUCC for different HRU were calibrated based on the data of calibration period (1960-1969) for every subbasin. Once the calibrated model was validated well in validation period (1970-1979), the parameters will not change. The comprehensive parameters can be calibrated according to observed streamflow of subbasin, while the different parameters in different slopes, soli type and LUCC could not calibrate individually. We added some details in uncertainty in simulated stream flow results (Line 462-464).

*2. How catchment characteristics like land use land cover, slope soil, drainage etc., have been adopted for analysis period i.e from 1980-2009, as these are very important for any hydrological phenomenon in addition to the meteorological parameters.*

Thank you for your suggestion. We have supplement information about it (line 203, 309 and 353-354). DEM was used to define the topography (such as elevation, slope and aspect) and delineate the watershed boundary. DEM and soil data remain constant. For period of 1980-2009, we used their measured and long-term daily meteorological data of the study area to drive the validated model. And to reduce influence of meteorological conditions; the 30-year (1980-2009)

values of streamflow were averaged respectively. For studies of impact of the observed LUCC on streamflow, the land use data of the 1980 and 2005 were used in the validated SWAT model and the DEM and soil data remained constant.

For studies of the impact of conversion of agricultural land to forests on streamflow, we designed the hydrological experiments using LUCC data of 1980 as (S1) the present land use, we design other four scenarios (Table 3) that (S2) 10%, (S3) 20%, (S4) 40% and (S5) 100% of the agricultural land was converted into Forest-Mixed (FRST) respectively. And all experiments carried out based on the same the DEM, soil data and meteorological conditions.

*3. Simulated stream flow results show a high coefficient of variation. Therefore, extent of uncertainty in simulated stream flow results may also be given before commenting on usefulness of study.*

This is a good suggestion. We supplemented the analysis for uncertainty in simulation (462-470).

First, the SWAT model could offer the comprehensive parameters for subbasin and detailed parameters for different HRU according to their slopes, soli type and LUCC. The comprehensive parameters were calibrated according to observed streamflow of subbasin, while the different parameters of HRU could not be calibrated individually. Second, the model could not tell the impact of short-duration rainfall on streamflow which has great effect on streamflow. In addition, watershed size, generalization and data accuracy all can lead to uncertainty in the simulations (Yin et al., 2017). To reduce the uncertainty of simulation influence, the 30-year (1980-2009) values of streamflow were averaged to analyze the impacts.

*4. China's Grain for Green project, represents the anthropogenic changes in hydrology, its impacts should be discussed and results of these impacts leading to conflict results.*

Thank you for your suggestion. We supplemented the related analysis in line 448-461. Although some researchers have urged a cessation on Grain for Green expansion on the Loess Plateau of China for it lead to annual streamflow of the Yellow River declining (Chen et al., 2015; Li, 2001), our modeling results suggest that forest recovery constructions have a little positive impact on both soil flow and base flow compensating reduced surface runoff, which leads to a slight increase in streamflow in the Wei River with mixed landscapes of Loess Plateau and earth-rock mountain. And rainfall patter also has great effect on streamflow, particularly the extremes rainfall, i.e., Lacombe et al. (2008) found no streamflow change was found for when the precipitation was larger than 40 mm. Results showed that the daily precipitation extremes seem to be consistent with the 7% increase per degree of warming (Allen and Ingram, 2002; Pall et al., 2007) and one-hour precipitation extremes increase twice as fast with rising temperatures as expected when daily mean temperatures exceed 12 $^{o}$C (Lenderink and Meijgaard, 2008; Westra,2014). The streamflow is the combined effects of LUCC (forestation, terraces, grass, and dam and so on) and climate changes. The impact of Grain for Green project on streamflow should be thoughtfully studied according to the characteristics of the basin.

*5. Latest article by J. Yin et al.2017 in HESS: Effects of land use/land cover and climate changes also shows the similar study results, this paper may also be referred here and results may also compared in discussion, which strengthens the quality of this article.*

Thank you for your recommendation. We have studied the reference thoughtfully and it offers a thought for revising and strengthening the quality of this manuscript. We have cited it as compared results (Line 322 and 468).

*In line 31: Model was calibrated for 10 years form 1960-69, validated for 1970-79 and even analysis is also a part of validation. Hence it is suggested to validate up to 2009. Large data used from1960-09 (50yrs).*

Thank you for your comments. Figure s1 shows the time-series graph of calculated streamflow vs. observed streamflow during 1980-2009 for hydrological stations. We can see the calculated streamflow matched well with the observed values during 1980s. The observed values were measured daily based on the actual LUCC, whereas the calculated streamflow was got based on LUCC of 1980. So Fig. s1 shows the calibrated SWAT model played well in our study area and the changing LUCC can affect streamflow gradually. To reduce influence of meteorological condition and isolate the impact of the LUCC on streamflow, 30-year average of the streamflow for forest and agricultural land were taken, respectively. Editor and Referees suggested one figure showing the observed and simulated streamflow was enough in previous version. We showed this result in Author's Response-RC2 in previous version and one figure (Fig.7) showing the observed and simulated streamflow was kept in the manuscript.

[Figure]

Figure s1 The time-series graphs of calculated vs. observed streamflow during 1980-2009 for
hydrological stations.

*In line 37: What is soil flow, how it differs from base flow? Explain. In my view baseflow and
soil flow were one and the same.*

Thank you for your comment. Yes, it is not necessary to part them in many studies.
SWAT partitions the streamflow which originates below the surface into Soil flow and base
flow (Arnold et al. 1993).

**Soil flow** is the lateral subsurface flow, or interflow. It is streamflow contribution which originates below the surface but above the zone where rocks are saturated with water. Lateral subsurface flow in the soil profile (0-2m) is calculated simultaneously with redistribution. A kinematic storage model is used to predict lateral flow in each soil layer. The model accounts for variation in conductivity, slope and soil water content.

**Base flow,** or return flow, is the volume of streamflow originates from groundwater. SWAT partitions groundwater into two aquifer systems: a shallow, unconfined aquifer which contributes return flow to streams within the watershed and a deep, confined aquifer which contributes return flow to streams outside the watershed.

We supplement the information in part "3.1 The SWAT model" (Line 218-221).

*Discussion: Reforestation reduced the surface runoff because of vegetation cover and increased streamflow may be because of increase in baseflow that depends on land topography, forestation might be on hillocks. The impact of Grain for Green project shows the conflict results i.e initially from 1969-1980 streamflow reduced and for present periods increasing. And also the Mountain rock regions, higher altitude variation including changing rainfall patter with intensive storms too leading to higher stream flow.*

Thank you for your suggestion. We supplemented the related analysis in line 448-461.

*Suggestions: One hydrological cycle represents for 30 years. In this study 50 years data used which too large. Study may be restricted to latest 30 years i.e 1980-2009. Calibration and validation may be carried out for the latest data as it represents' the real conditions specially rainfall distribution pattern.*

Thank you for your suggestion. For period of 1980-2009, we used their measured and long-term daily meteorological data of the study area to drive the validated model. And To reduce the uncertainty of simulation influence, the 30-year (1980-2009) values of streamflow were averaged to analyze the impacts.

*Finally this article, Impact of LUCC on Streamflow Based on the SWAT Model over the Wei River Basin on the Loess Plateau of China by Hong Wang and Fubao Sun is worth of publishing in HESS with minor revisions as suggested.*

Thank you for your useful suggestions and constructive comments on our manuscript.

**Part 2. A marked-up manuscript version**

[revised manuscript text omitted]